# Augmented microglial endoplasmic reticulum-mitochondria contacts mediate depression-like behavior in mice induced by chronic social defeat stress

Extracellular ATP (eATP) signaling through the P2X7 receptor pathway is widely believed to trigger NLRP3 inflammasome assembly in microglia, potentially contributing to depression. However, the cellular stress responses of microglia to both eATP and stress itself remain largely unexplored. Mitochondria-associated membranes (MAMs) is a platform facilitating calcium transport between the endoplasmic reticulum (ER) and mitochondria, regulating ER stress responses and mitochondrial homeostasis. This study aims to investigate how MAMs influence microglial reaction and their involvement in the development of depression-like symptoms in response to chronic social defeat stress (CSDS). CSDS induced ER stress, MAMs' modifications, mitochondrial damage, and the formation of the IP3R3-GRP75-VDAC1 complex at the ER-mitochondria interface in hippocampal microglia, all concomitant with depression-like behaviors. Additionally, exposing microglia to eATP to mimic CSDS conditions resulted in analogous outcomes. Furthermore, knocking down GRP75 in BV2 cells impeded ER-mitochondria contact, calcium transfer, ER stress, mitochondrial damage, mitochondrial superoxide production, and NLRP3 inflammasome aggregation induced by eATP. In addition, reduced GRP75 expression in microglia of Cx3cr1$^{CreER/+}$Hspa9$^{f/+}$ mice lead to reduce depressive behaviors, decreased NLRP3 inflammasome aggregation, and fewer ER-mitochondria contacts in hippocampal microglia during CSDS. Here, we show the role of MAMs, particularly the formation of a tripartite complex involving IP3R3, GRP75, and VDAC1 within MAMs, in facilitating communication between the ER and mitochondria in microglia, thereby contributing to the development of depression-like phenotypes in male mice.

Depression is a prevalent mental illness with significant morbidity and mortality. It's widely acknowledged that elevated levels of external stressors can contribute to the development of depression[1–3]. Previous findings have demonstrated that extracellular ATP (eATP) acts as a stress signal via P2X7 receptors (P2X7Rs) to induce NLRP3 inflammasome assembly[4,5], an important event in the pro-inflammatory status of hippocampal microglia in depression development[6]. Notably, pharmacological studies suggest that P2X7Rs

✉ e-mail: yujin@shmu.edu.cn

require relatively high concentrations of ATP, with full activation occurring at 1 mM ATP[7,8]. Despite these insights, the specific intracellular mechanisms at play remain elusive.

When cells encounter stress, they activate intracellular processes to maintain their functions and overall microenvironment, striving to uphold the body's internal balance. However, these responses can sometimes lead to maladaptive changes that contribute to the development of various diseases[9]. Recent research has suggested that the malfunctioning of cellular organelles, such as the endoplasmic reticulum (ER) and mitochondria, may play a significant role in the pathophysiological changes that occur in response to stress[10]. Intriguingly, clinical studies have uncovered potential links between ER stress, mitochondrial dysfunction, and psychiatric disorders, specifically major depression[11,12].

Recent evidence underscores the significance of contact sites between the ER and mitochondria, known as Mitochondria-Associated ER Membranes (MAMs). These structures regulate a range of vital physiological functions, including calcium signaling, mitochondrial dynamics, autophagy, and cell death[13,14]. Among these functions, a tripartite complex composed of the inositol 1,4,5-triphosphate receptor 3(IP3R3), the voltage-dependent anion channel 1 (VDAC1), and the MAMs tethering protein glucose-regulated protein 75 (GRP75) has emerged as a key player in ER-mitochondrial $Ca^{2+}$ signaling through physical interactions between these organelles[15]. In addition, MAMs may play a central role in conveying ER stress to mitochondria[16], simultaneously, mitochondrial DNA (mtDNA) and reactive oxygen species (ROS) represent major triggers for NLRP3 inflammasome assembly[17,18].

Therefore, the primary goal of this study is to uncover the role of MAMs in ER-mitochondria communication within microglia and determine if these processes are involved in eATP-induced microglial response, contributing to depression development. The results indicate that the formation of a tripartite complex involving IP3R3, GRP75, and VDAC1 within MAMs strengthens the connections between the ER and mitochondria in microglia. This process contributes to the microglial response triggered by eATP and the development of depression-like behaviors induced by CSDS in male mice.

## Results

### CSDS induces ER stress, structural and functional changes in MAMs, and mitochondrial damage in hippocampal microglia

Our previous studies have suggested that hippocampal microglia might play a key role in chronic unpredictable mild stress (CUMS)-induced depression-like behaviors[6]. In this study, we further confirm the cellular stress responses of microglia to stress stimulus. We used CD11b+ magnetic beads to separate hippocampal microglia from CSDS and control mice, followed by RNA sequencing (RNA-seq) analysis (Fig. 1a). Based on the RNA-seq findings, chronic social defeat stress (CSDS) demonstrated a substantial upregulation of unfolded protein response (UPR)-related pathways, including *Atf4*, *Caspase12*, *Atf6*, and others, as well as mitoptosis pathways, such as *Sod1*, *Cyc1*, *Vdac1*, and others, in comparison to control conditions (Fig. 1b). And an increase in the expression of Caspase12 and VDAC1 was observed in microglia of the hippocampus after CSDS (Supplementary Fig. 1a, b). Gene Ontology (GO) analyses showed differences in genes encoding organelles, stress responses (ER-unfolded protein response) and cellular processes between control and CSDS mice (Fig. 1c, d). Transmission electron microscopy (TEM) revealed that CSDS led to the expansion of the ER and swelling of mitochondria in microglia (Fig. 1e), indicating that activated microglia in the hippocampus of CSDS mice may experience ER stress and mitochondrial damage. Additionally, we investigated whether CSDS also affected MAMs, the dynamic platform connecting ER and mitochondria in hippocampal microglia. Immuno-electron microscopy (immuno-EM) and TEM results showed that CSDS increased the association between ER and mitochondria in

hippocampal microglia (Fig. 1e, f). Furthermore, in situ Proximity Ligation Assay (PLA) analysis revealed a significant increase in the IP3R3-GRP75-VDAC1 complex within microglia of the hippocampus (Fig. 1g). This complex has been previously validated as being situated at the interface between the ER and mitochondria, playing an essential role in facilitating mitochondrial calcium uptake. The cellular stress responses observed were concurrent with morphological alterations in hippocampal microglia after exposure to CSDS. These alternations were evidenced by the heightened fluorescence intensity of Iba-1 (Supplementary Fig. 1c), an increase in branch number, and a decrease in branch length (Supplementary Fig. 1d) in DG region, accompanied by the manifestation of depression-like behaviors in mice (Supplementary Fig. 1e), including social avoidance, behavioral despair, and anhedonia. A significant increase in iba-1 expression was also found in the CA1 and mPFC regions of the brain caused by CSDS (Supplementary Fig. 1h, i). Furthermore, RNA-seq analysis of hippocampal microglia (Supplementary Fig. 1g) and RT-PCR of hippocampal tissues (Supplementary Fig. 1f) revealed augmented expression levels of several surface receptors in microglia, notably CD68, CX3CR1, MHC II, and CD40, after exposure to CSDS, except for Arg1. In summary, these results suggest that CSDS upregulates the sensitivity of hippocampal microglia and leads to ER stress, increased ER-mitochondria interaction, and mitochondrial dysfunction within microglia, along with the development of depressive-like behaviors in mice.

### Extracellular ATP (eATP) leads to ER stress, alterations in MAMs, and mitochondria damage in microglia

Subsequently, we validated that the eATP-P2X7 receptor signaling pathway orchestrates microglial responses induced by chronic social defeat stress (CSDS), consequently contributing to the emergence of depressive-like behaviors (Supplementary Fig. 2), consistent with earlier investigations[4,5]. This rationale guided our utilization of ATP as an extracellular danger signal to replicate the microenvironment induced by CSDS, enabling an in-depth exploration of the intricate mechanisms governing stress responses in microglia (primary microglia, Fig. 2 and BV2 cells, Supplementary Fig. 3). The results demonstrated that after 2 h of ATP (1 mM) or 12 h of ER stress inducer Thapsigargin (TG, 1 μM, used as a positive control, which raises cytosolic calcium concentration by blocking the ability of the cell to pump calcium into the sarcoplasmic reticulum and ER) treatment resulted in an augmentation of ER-mitochondria contacts, as evidenced by transmission electron microscopy experiments (Fig. 2a and Supplementary Fig. 3a) and corroborated by live cell fluorescence marker staining (Supplementary Fig. 3b), activation of the PERK-eIF-2α pathway (Supplementary Fig. 3c), mitochondrial dysfunction, including a loss of mitochondrial membrane potential (ΔΨm) (indicated by a decreased ratio of JC-1 aggregates to JC-1 monomers, Supplementary Fig. 3d), elevated oxidative stress (as evidenced by increased ROS and mitoSOX levels, Fig. 2b, c), enhanced mitophagy (as indicated by greater co-localization of MitoTracker Green and LysoTracker Red, Fig. 2e), and increased assembly of the NLRP3 inflammasome (Supplementary Fig. 3e). Additionally, ATP stimulation increased the assembly of the IP3R3-GRP75-VDAC1 complex (Fig. 2e and Supplementary Fig. 3f), accompanied by elevated cytoplasmic (Fig. 2f) and mitochondrial $Ca^{2+}$ levels (Fig. 2g, lower panel), and reduced ER $Ca^{2+}$ levels (Fig. 2g, upper panel). These findings collectively suggest that eATP induces ER stress, mitochondrial dysfunction, and ER-mitochondrial tethering, potentially facilitating $Ca^{2+}$ transfer from the ER to mitochondria in microglia.

### IP3R-GRP75-VDAC complex participates in the changes in MAMs structure and function induced by eATP

The GRP75-IP3R3-VDAC complex in MAMs plays an important role in mediating the tethering between the ER and mitochondria. This process is vital for MAMs formation and facilitates the transfer of $Ca^{2+}$

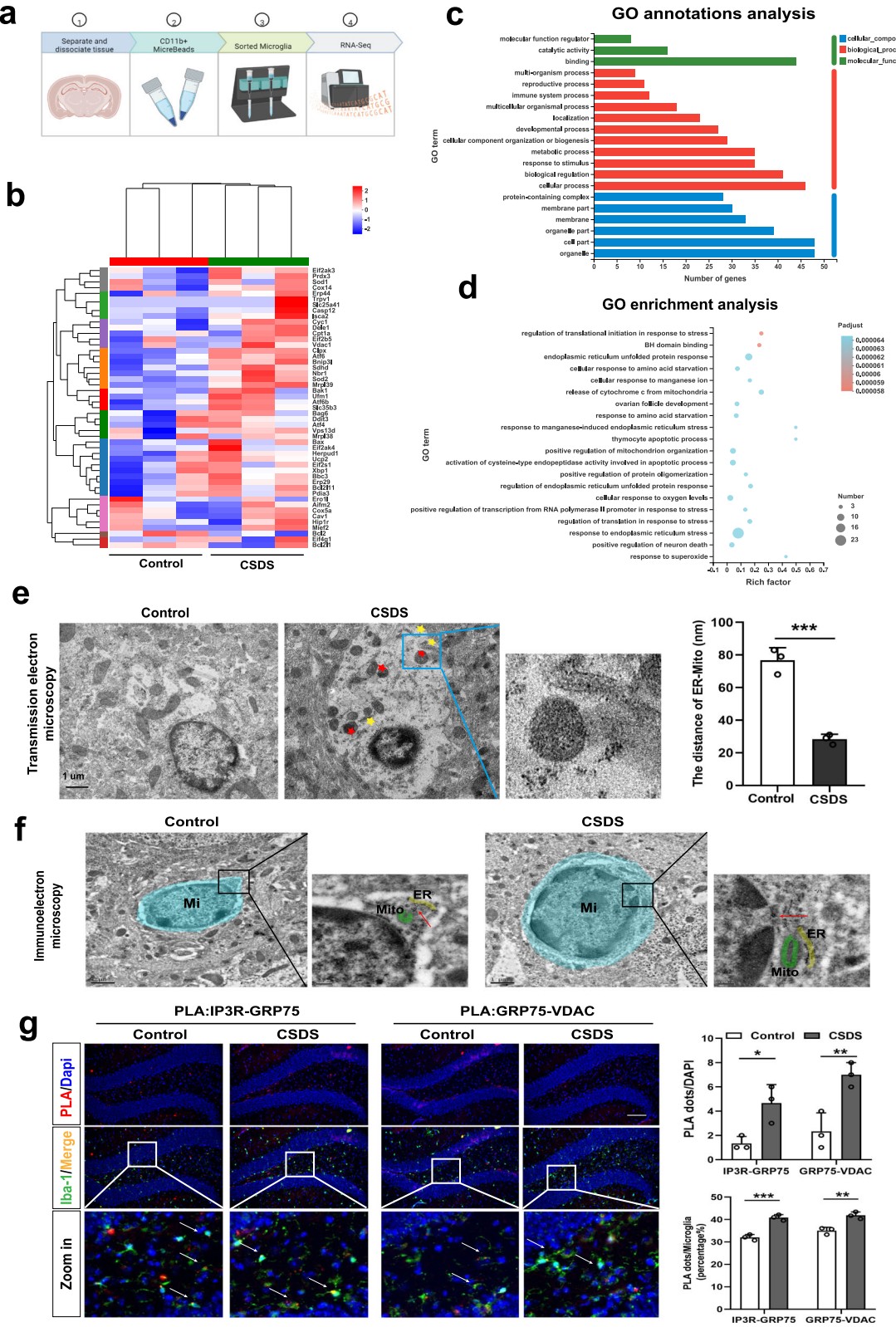

from the ER to mitochondria in eukaryotic cells, especially immune system cells[19]. Following this, we explored the potential influence of GRP75 knockdown on the structural and functional alterations in MAMs induced by eATP in microglia. Knocking down GRP75 in BV2 cells (Fig. 3a) inhibited the augmentation of ER-mitochondrial contacts, evidenced by the absence of a further reduction in the distance between these organelles (Fig. 3b). Additionally, GRP75 knockdown prevented the transfer of ER $Ca^{2+}$ to mitochondria induced by eATP

(Fig. 3c). Furthermore, it mitigated eATP-induced ER stress (Fig. 3d), mitochondrial damage (Fig. 3e), generation of mitochondrial superoxide (Fig. 3f, g), and aggregation of NLRP3 inflammasomes and Caspase-1 or ASC protein level (Fig. 3h). These results substantiate our hypothesis that the IP3R3-GRP75-VDAC1 tripartite complex facilitates the tethering of the ER and mitochondria, along with the transfer of $Ca^{2+}$ from the ER to the mitochondria. Both processes are important for the cellular stress response elicited by eATP in microglia.

**Fig. 1 | Chronic stress induces ER stress, structural and functional changes in MAMs, and mitochondrial damage in hippocampal microglia. a** Flow chart for isolating microglia via CD11b$^+$ magnetic beads from adult mouse hippocampus and subsequent RNA-Sequencing of microglia. Figure 1a was created with BioRender.com released under a Creative Commons Attribution-Non Commercial-No Derivs 4.0 International license. **b** Heatmap showing the different expressed genes in microglia isolated from hippocampus of the Control (n = 15) and chronic social defeat stress (CSDS) mice (n = 15), as determined by RNA-seq data. Ten hippocampi from 5 mice were combined to one sample. **c, d** Gene Ontology (GO) annotations and enrichment of the DEGs between Control and CSDS. Using Fisher's exact test, when the corrected $P$ value (Padjust) is less than 0.05, it is considered that there is a significant enrichment of this GO function. **e** Representative Transmission Electronic Microscope (TEM) images (left panel) and quantitative analysis of the distance between mitochondria and ER (right panel) of microglia in the hippocampus of control group and CSDS group mice. Scale bar: 1.0 μm; Student's $t$ test (P = 0.0006). Data are expressed as mean ± SEM (n = 3 per group). ***$p < 0.001$. **f** Immunoelectron microscope images with higher magnification of the inset showing the intimate proximity between mitochondria (M) and ER in a gold labeled microglia (Mi) (Red arrow: dense black dots) located in the hippocampal DG from Control and CSDS mice. **g** Representative images of PLA targeting IP3R3-GRP75 or GRP75-VDAC1 interactions in the hippocampal DG (left panel) and Quantification of the PLA red fluorescent dots (Top line, Interaction, F$_{(1, 8)}$ = 0.8889, P = 0.3734) and the percentage of PLA dots in microglia (Bottom line, Interaction, F$_{(1, 8)}$ = 1.575, P = 0.2449) were performed using Image J (right panel). Scale bar: 20 μm; Microglia (Iba-1, green), Nuclei (DAPI, blue). Two-way ANOVA with Sidak's multiple comparisons test. Data are expressed as mean ± SEM (n = 3 mouse brain slices). *$p < 0.05$, **$p < 0.01$, ***$p < 0.001$. Source data are provided as a Source Data file.

## Conditional knockdown of GRP75 in hippocampal microglia attenuates CSDS-induced depression-like behaviors

We employed conditioned microglia-specific *Hspa9* knockout mice (Cx3cr1$^{CreER/+}$/Hspa9$^{f/+}$, Fig. 4a and Supplementary Fig. 4) to confirm whether GRP75 (*Hspa9*) is involved in CSDS-induced depression-like behaviors. Following a 5-day tamoxifen treatment (intraperitoneal injection, i.p.), the expression of GRP75 protein in microglia in the hippocampus or mPFC brain area decreased by approximately 30–40%. (Fig. 4b and Supplementary Fig. 4b). To assess the impact of specific GRP75 knockdown in microglia on CSDS-induced depression-like behavior, tamoxifen administration coincided with CSDS modeling (Fig. 4c). Compared to Cx3cr1$^{CreER/+}$ mice, Cx3cr1$^{CreER/+}$/Hspa9$^{f/+}$ mice displayed reduced depressive behaviors after exposure to CSDS and tamoxifen injection, exhibiting decreased immobility time and a stronger preference for sucrose solution over distilled water (Fig. 4d). Meanwhile, knocking down GRP75 does not affect the CD68 expression within microglia in the hippocampus caused by CSDS (Supplementary Fig. 4c). Additionally specific GRP75 knockdown in microglia attenuated the PERK-eIF-2α pathway protein expression (Fig. 4e), mitochondrial dysfunction (Fig. 4f), the aggregation of NLRP3 inflammasomes (Fig. 4g) and the enrichment of ER-mitochondria contacts in hippocampal microglia triggered by CSDS (Fig. 4h). However, the above results were not observed in unmodified mice (Supplementary Fig. 4d–g). In summary, these findings underscore the role of GPR75 in mediating CSDS-induced depression-like behavior and cellular stress responses of microglia.

## P2X7 receptors mediate CSDS-induced alterations in MAMs within hippocampal microglia

Finally, we explored the involvement of P2X7R in the formation of the IP3R3-GRP75-VDAC1 complex within microglia in response to eATP, as well as its impact on the changes in MAMs within hippocampal microglia and the subsequent development of depression-like behaviors triggered by CSDS. Our findings, based on primary microglia from *P2X7R$^{-/-}$* mice, demonstrated that the deletion of P2X7R prevented the enrichment of ER-mitochondria contacts and the formation of the IP3R3-GRP75-VDAC1 complex (Fig. 5a, b) induced by a high concentration of eATP. Similar results were obtained using siRNA-treated BV2 cells (Supplementary Fig. 5a, b). Additionally, the structural changes in MAMs in hippocampal microglia (Fig. 5c) and depression-like behaviors (Supplementary Fig. 2e) caused by exposure to CSDS were attenuated in *P2X7R$^{-/-}$* mice.

In summary, our results provide helpful insights into the role of these cellular stress responses in depression-like behavior (Graphical abstract, Fig. 5d), and suggest that the formation of the IP3R3-GRP75-VDAC1 complex and the changes in the structure and function of MAMs may be significant cellular responses in microglia triggered by eATP-P2X7Rs signaling, potentially contributing to CSDS-induced microglia responses and the development of depression-like phenotypes.

## Discussion

In this study, we have demonstrated that ER-mitochondria contact sites function as central hubs for cellular stress responses, contributing to mood disorders like depression-like behavior by inducing both ER stress and mitochondrial damage. Additionally, the activation of the eATP-P2X7R signaling pathway disrupts the structural and functional equilibrium of ER-mitochondria contacts, consequently triggering NLRP3 inflammasomes assembly in microglia and further contributing to the development of depression-like behavior. Moreover, the formation of a triple complex IP3R3-GRP75-VDAC1 at the ER-mitochondria contact sites is accompanied by an increase in both ER-mitochondria contacts and the transfer of calcium ions (Ca$^{2+}$) to the mitochondria. These processes play an important role in transmitting extracellular stress signals, such as eATP, ultimately leading to mitochondrial damage and the activation of NLRP3, which is implicated in the stress-induced development of depression-like behavior. Our findings offer an insight into the role of intracellular stress in the biology of depression, with a particular emphasis on the disruption of structural and functional homeostasis within ER-mitochondria contacts in microglia.

Our present experiments provide further support for earlier research findings[4,5], which demonstrated that extracellular ATP (eATP) acts as a potent stimulus for the release of mature IL-1β from microglia, thus serving as a mediator for stress-induced depression. Moreover, previous studies[4,6] have established the activation of the NLRP3 inflammasome as the most potent downstream effector of P2X7R, facilitating the cleavage of caspase-1 and subsequently the maturation of IL-1β. The current in vitro results offer additional insights into the intracellular mechanisms governing the activation of the NLRP3 inflammasome triggered by ATP via P2X7R in microglia, aligning with previous research findings in immune cells[20].

Recent findings have linked immune-related genes encoding P2X7R to severe depression and bipolar disorder (BD). In line with our previous investigation[5] and the current study, the absence of P2X7R failed to elicit significant depressive-like behavior in either the CUMS[21] or CSDS models. Furthermore, several clinical studies have also established genetic associations between depression and *P2RX7*. Bioinformatics analysis of diseases such as MDD and BD have identified a shared positive locus for a gene in the 12q region of the chromosome, which encompasses the gene encoding P2X7R[22]. A large-scale genome-wide association study integrating large-scale human genetic research and clinical drug targets, as well as whole proteome MR analysis, has highlighted an association between *P2RX7* gene and depression[23]. Furthermore, a case study report has confirmed that BD is associated with alleles between markers rs2230912 (*P2RX7*-E13A, G allele, P = 0.043) and NBG6 (P = 0.010)[24]. These studies, in conjunction with our own investigations, underscore the close relationship between the *P2RX7* gene and depression.

As previously mentioned, the NLRP3 inflammasome plays an important role in immune cells, contributing to both innate and

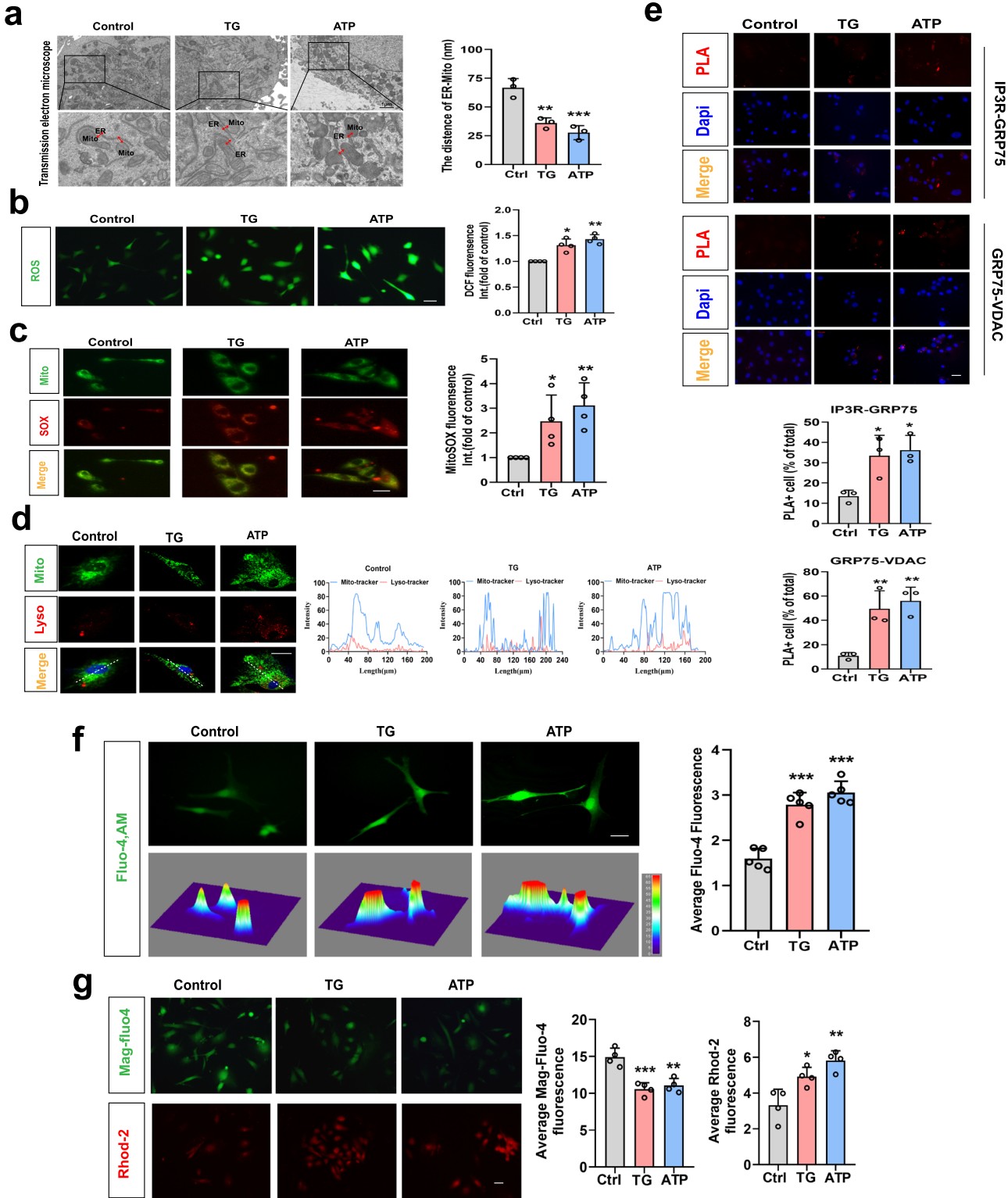

adaptive immune responses. This is primarily because NLRP3 can respond to a wide range of structurally unrelated pathogen-associated molecular patterns (PAMPs)[25]. Recently, it has become evident that damage-associated molecular patterns (DAMPs) can also trigger the activation of the NLRP3 inflammasome[26]. Consequently, abnormal NLRP3 inflammasome activation has been implicated in various human diseases, including diabetes mellitus[27], depression[28], and multiple sclerosis[29]. Nevertheless, the precise mechanism governing NLRP3 activation remains elusive.

Considering the vast number and structural diversity of NLRP3 activators, it appears that NLRP3 does not directly engage with PAMPs or DAMPs on a physical level. Instead, it appears to sense common intracellular events induced by these activators. Over the past decade, numerous intracellular events have been proposed as triggers for NLRP3 inflammasome activation. Various signals, such as mtROS production, mtDNA release, and cardiolipin externalization, have been suggested as mediators of NLRP3 inflammasome activation through mitochondrial dysfunction[30–33]. Our findings affirm the concept that, akin to peripheral

**Fig. 2 | Extracellular ATP (eATP) leads to ER stress, alterations in MAMs, and mitochondria damage in microglia. a** Representative TEM images of ER-mitochondria contact in primary microglia untreated (Ctrl) or treated with Thapsigargin (TG) or Adenosine Triphosphate (ATP, left panel) and the quantitative analysis of the distance between mitochondria and the ER (right panel, n = 3 mitochondria in 3 fields per condition; P = 0.0020, P = 0.0006). Scale bar: 1 μm. **b** Representative photographs of DCF fluorescence in the experimental groups (left panel) and Relative DCF fluorescence was quantitatively analyzed (right panel, n = 4 per condition; P = 0.0012, P = 0.0001). Scale bar: 20 μm. **c** Representative images of MitoSOX (red) and Mito-tracker (green) fluorescence in primary microglia (left panel). Relative MitoSOX fluorescence was quantitatively analyzed (right panel, n = 4 per condition; P = 0.0159, P = 0.0050). Scale bar: 20 μm. **d** Representative confocal images (left panel) and the fluorescence intensity at the line of the arrow in left images (right panel) of intracellular localization of lysosome (Lyso-Tracker, red), mitochondria (Mito-Tracker, green) and nucleus (Hoechst, blue). Scale bar: 20 μm. **e** Representative images of proximity ligation assay (PLA) targeting IP3R3-GRP75 or GRP75-VDAC1 interactions (above panel) and quantification of the PLA red fluorescent dots in microglia (below panel, n = 3 batches of primary microglia; P = 0.0283, P = 0.0164; P = 0.0090, P = 0.0042). Scale bar: 20 μm. **f** Representative fluorescence images, 3D thermograms (left panel) and quantitative analysis (right panel, n = 5 per condition; P < 0.0001, P < 0.0001) of Fluo-4 in microglia. Scale bar: 20 μm. **g** Representative fluorescence images (left panel) and quantitative analysis (right panel, n = 4 per condition; P = 0.0005; P = 0.0012) of Mag-Fluo-4 and Rhod-2 in microglia. Scale bar: 20 μm. One-way ANOVA with Dunnett's multiple comparisons test. Data are expressed as mean ± SEM *p < 0.05, **p < 0.01, ***p < 0.001, vs. Ctrl. Source data are provided as a Source Data file.

immune cells, intracellular mitochondrial dysfunction acts as an upstream trigger for NLRP3 inflammasome activation within microglia and is linked to ATP-induced NLRP3 inflammasome activation.

In addition to mitochondrial dysfunction, ion fluxes, specifically $K^+$ efflux, $Ca^{2+}$ mobilization, and $Cl^-$ efflux, have also been proposed as key upstream events in the activation of the NLRP3 inflammasome[34–36]. Of particular interest, it has been observed that $Ca^{2+}$ influx triggered by P2X7R and other $Ca^{2+}$ channels serve as a stimulus for the mobilization of $Ca^{2+}$ from the endoplasmic reticulum (ER) storage in immune cells[18,34,37]. Simultaneously, $Ca^{2+}$ depletion or continuous release from the ER can lead to ER stress and the initiation of the unfolded protein response (UPR)[38].

Given that mitochondrial dysfunction, involving mtROS generation, cardiolipin externalization, and mtDNA release, has been implicated as triggers of NLRP3 inflammasome activation, and concurrently, calcium release from the ER induces mitochondrial calcium overload[39,40], leading to mitochondrial damage, it is plausible that calcium release in the ER can subsequently trigger the activation of the NLRP3 inflammasome.

It is likely that the cytoplasmic $Ca^{2+}$ concentration does not reach the threshold for mitochondrial damage and NLRP3 activation under conditions of $Ca^{2+}$ mobilization[41,42]. Alternatively, the transfer of $Ca^{2+}$ from the ER to mitochondria through the presence of MAMs may elevate $Ca^{2+}$ mobilization to a level that causes mitochondrial damage. In essence, MAMs serve as a fundamental platform for various cellular functions, including the transfer of $Ca^{2+}$ from the ER to mitochondria.

Within MAMs, the IP3R3- GRP75 -VDAC1 complex facilitates $Ca^{2+}$ transfer from the ER to mitochondria[14,43] and plays a role in the physical interaction between the ER and mitochondria[44,45]. This aligns with our results, which demonstrate that the knockdown of GRP75 reduces ER-mitochondria coupling, diminishes $Ca^{2+}$ transfer from the ER to mitochondria, and prevents the transmission of ER stress to mitochondria. Simultaneously, it abolishes the impact of ATP on mitochondrial damage and subsequent NLRP3 activation in microglia. This finding underscores the significance of the IP3R3-GRP75-VDAC1 complex in the cellular stress responses induced by ATP in microglia.

As mentioned earlier, MAMs play a significant role in facilitating communication between ER stress and mitochondrial damage, both of which are crucial cellular stress responses that mediate the proinflammatory effects of ATP on microglia. Conversely, these cellular stress responses have not only been associated with the maintenance of systemic homeostasis but also with an inability to adapt to stress, potentially contributing to stress-related disorders like depression[9].

Support for the connection between systemic ER stress and stress-related major depressive disorder (MDD) has been reported by Brown et al., who observed up-regulated expression of ER stress-related proteins in suicidal MDD patients compared to those with other causes[46]. This association between ER stress and stress-related MDD is further supported by another study demonstrating increased expression of unfolded protein response (UPR)-related genes, such as *BIP*, *CHOP*, and *XBP1s*, in leukocytes[11].

Furthermore, Gardner et al. reported that 68% of depressive patients exhibited mitochondrial DNA (mtDNA) deletions, in contrast to 36% of control subjects[47]. Additionally, a recent genetic study provided further evidence for the connection between mitochondrial dysfunction and depression. It implicated several mitochondrial genes, including *TOMM40* and *MAO* genes, which encode the translocase of the outer mitochondrial membrane pore subunit and mitochondrial monoamine oxidase (MAO) isozymes A and B, respectively, in depression[48].

Our present results reinforce the link between ER stress, mitochondrial damage, and depression-like behavior. They suggest that social stress leads to extensive ER stress and mitochondrial damage in the hippocampus, consequently inducing depression-like phenotypes.

Our observations may have a broader impact on our understanding of depression. Increased MAMs function and enhanced ER-mitochondria interaction have also been observed in neurodegenerative diseases such as gangliosidosis and Alzheimer's disease. Notably, Alzheimer's disease is closely associated with the IP3R3-GRP75-VDAC1 complex in the brain and shares numerous molecular features, including ER stress, organelle dysfunction, and mitochondrial dysfunction[44,49,50]. Interestingly, scientists have noted that changes in behavior, including depression, may precede memory loss in seniors who eventually develop Alzheimer's disease. In other words, depression could potentially serve as a risk factor or early symptom of Alzheimer's disease[51,52]. Therefore, we suggest that therapies aimed at restoring the proper function of the ER-mitochondria interface could potentially help in treating a range of affective disorders with similar underlying mechanisms.

## Methods
### Animals
Eight-week-old C57BL/6J mice and seven-month-old CD1 mice were obtained from Slack Laboratory Animal Co., Ltd. (Shanghai, China), $P2X7R^{-/-}$ mice (Strain #:005576) and Cx3cr1CreER mice (Strain #:021160) were purchased from the Jackson Laboratory (Maine, U.S.A). Conditional microglia-specific Hspa9 knockout mice were generated using the Cre/LoxP system. Hspa9$^{flox/+}$ mice (Strain NO. T008726) under C57BL/6J background were purchased from GemPharmatech LLC. (Nanjing, China). All animals were habituated in 12 h light/dark cycle and allowed free access to food and water under conditions of controlled humidity and temperature (24 ± 0.5 °C). All experiments were carried out in accordance with the National Institutes of Health Guide for the Care and Use of Laboratory Animals and approved by the Experimental Animal Ethics Committee of Shanghai Medical College, Fudan University, Shanghai, China (20160225-071)."

### Cell lines and siRNA transfection
The BV2 murine microglial cell line was acquired from the Cell Bank of the Chinese Academy of Sciences (Shanghai, China). Primary microglia were extracted from hippocampus of fetal C57BL/6 mice. Primary $P2X7R^{-/-}$ microglia were extracted from hippocampus of newborn $P2X7R^{-/-}$ mice within 24 h. Cells were cultured in Dulbecco's modified Eagle's medium (DMEM, Gibco, New York, USA) with 10% fetal bovine serum (FBS, Gibco,

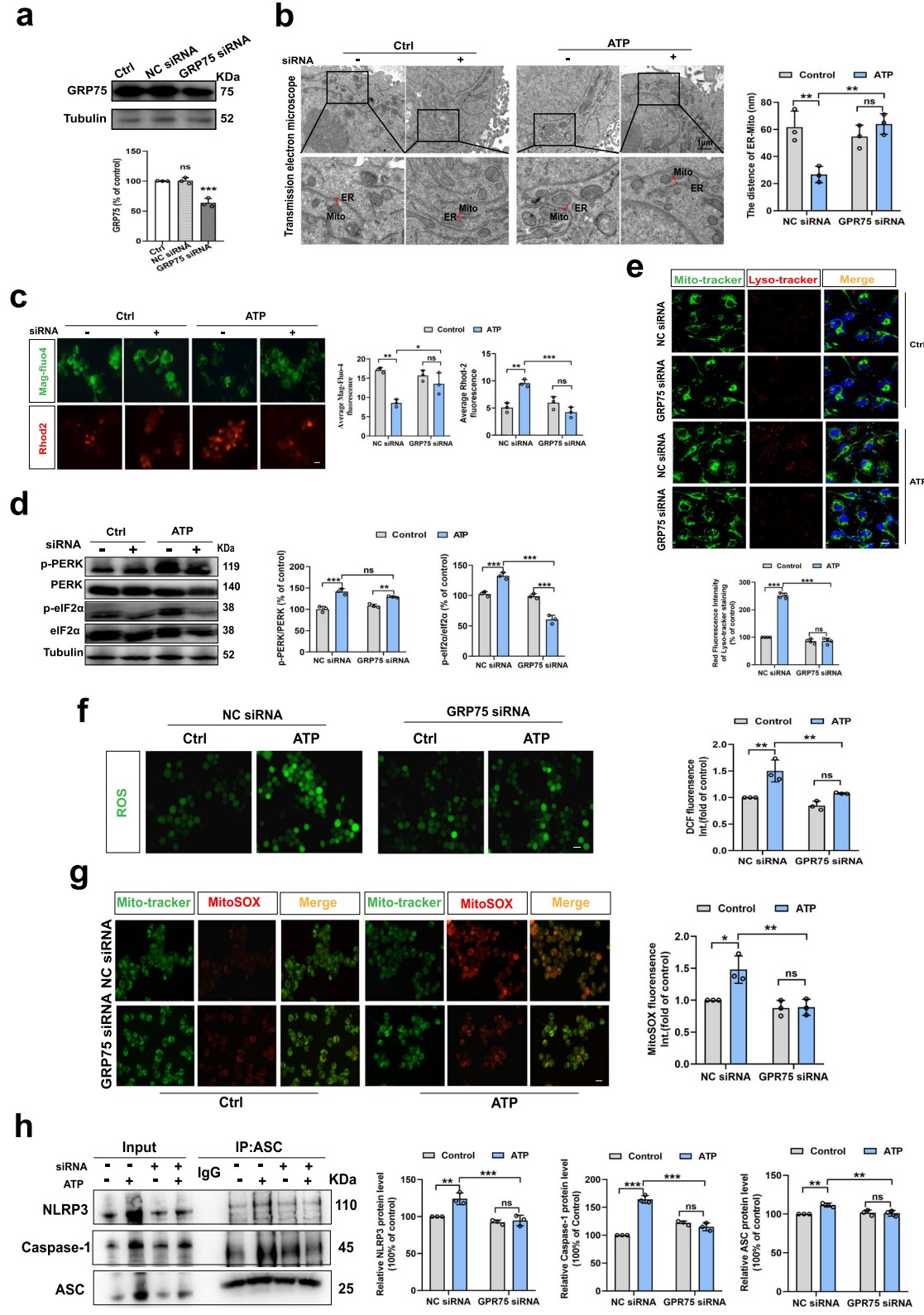

New York, USA) and penicillin-streptomycin (10 000 U/mL, Gibco, New York, USA), and then maintained at 37 °C with 5% $CO_2$.

### Chronic social defeat stress (CSDS) procedure and behavioral testing

CSDS procedure was performed as described in a previous protocol[53]. One C57BL/6 mouse (intruder) was placed in the home cage of a CD1 mouse (aggressor) and had physical interaction for 5–10 min for a total of 10 days. After contact, the intruder and aggressor mouse were separated by plexiglass partition with small holes to maintain stress exposure through visual, auditory, and olfactory stimulation for 24 h. It was ensured that the C57 mice were exposed to a new, never previously seen, CD1 aggressor every day for ten days.

**Social interaction test (SIT).** Before the test, the mice were placed in the behavior room and allowed to adapt to the indoor conditions for at

**Fig. 3 | IP3R3-GRP75-VDAC1 complex participates in the changes in MAMs structure and function induced by eATP. a** Western blot analysis of GRP75 proteins expression in BV2 cells subjected to transfection of GRP75 siRNA or negative control (NC) siRNA (n = 3 batches of transfected cell samples; P = 0.9897; P = 0.0003). **b** Representative TEM images of ER-mitochondria contact (left panel) and the quantitative analysis of the distance (right panel, n = 3 mitochondria in 3 fields per condition; Interaction, $F_{(1, 8)}$ = 19.14, P = 0.0024). Scale bar: 1 μm. **c** Representative fluorescence images (left panel) and quantitative analysis (right panel, n = 3 per condition; Interaction, $F_{(1, 8)}$ = 11.11, P = 0.0103; Interaction, $F_{(1, 8)}$ = 33.60, P = 0.0004) of Mag-Fluo-4 and Rhod-2 in BV2 cells. Scale bar: 20 μm. **d** Western blot analysis of PERK and elf2α phosphorylation proteins (n = 3 batches of transfected cell samples; Interaction, $F_{(1, 8)}$ = 10.79, P = 0.0111; Interaction, $F_{(1, 8)}$ = 124.7, P < 0.0001). **e** Representative confocal images (above panel) of the intracellular localization of lysosome (Lyso-Tracker, red), mitochondria (Mito-Tracker, green) and nucleus (Hoechst, blue) in BV2 cells and the statistical analysis (below panel, n = 4; Interaction, $F_{(1, 8)}$ = 77.58, P < 0.0001) of the fluorescence intensity. Scale bar: 20 μm. **f** Representative photographs of DCF fluorescence (left panel) and Relative DCF fluorescence was quantitatively analyzed (right panel, n = 3; Interaction, $F_{(1, 8)}$ = 4.353, P = 0.0704). Scale bar: 20 μm. **g** Representative images of MitoSOX (red) and Mito-tracker (green) fluorescence in BV2 cells (left panel). And relative MitoSOX fluorescence was quantitatively analyzed (right panel, Interaction, $F_{(1, 8)}$ = 8.536, P = 0.0192). Scale bar: 20 μm. **h** Representative blots of co-immunoprecipitation (IP, left panel) and quantification of the NLRP3, Caspase-1 and ASC protein changes (n = 3 batches of transfected cell samples; Interaction, $F_{(1, 8)}$ = 11.48, P = 0.0095; Interaction, $F_{(1, 8)}$ = 140.9, P < 0.0001; Interaction, $F_{(1, 8)}$ = 14.41, P = 0.0053). One-way or Two-way ANOVA with Tukey's multiple comparisons test. Data are expressed as mean ± SEM. ns, $p > 0.05$, $*p < 0.05$, $**p < 0.01$, $***p < 0.001$. Source data are provided as a Source Data file.

least 1 h. After testing each mouse, the test instrument was wiped with 75% alcohol to prevent odor interference. The SIT was performed on the first day after CSDS modeling using a new CD1 mouse. The experiment lasted for five min. The CD1 mice were not placed in the first 2.5 min and were allowed to explore freely in the field (44 × 44 cm), where a rectangular container (10 × 6 cm) was placed. CD1 mice were placed in the container 2.5 min later and counted in the "interaction area" (14 × 26 cm). The statistical formula for the social interaction ratio was as follows: time in the interaction area when there was a CD1 target mouse/time in the interaction area when there was no CD1 target mouse × 100%.

**Open field test (OFT).** The test mice were placed in a plastic box (50 cm × 50 cm × 60 cm), and the square area in the middle was 25 cm × 25 cm, allowing free exploration of the surroundings. The movement track of each animal within 5 min was recorded, and the total distance of mouse movement was recorded using the EthoVision XT software.

**Forced swimming test (FST).** The test mice were placed in a plexiglass cylinder (30 cm high and 15 cm in diameter), water was poured into the lower end (15 cm high and 25 ± 1 °C), and the mice swam for 5 min. Immobility was defined as the maintenance of the head above water without any movement. EthoVision XT software was used to evaluate the data.

**Sucrose preference test (SPT).** Mice were acclimated to two identical bottles for 24 h (20 mL, first 2% sucrose, second pure water), and the two bottles were switched after 12 h. After 24 h of fasting and water deprivation, a total of 8 h of testing was conducted, in which the bottle position was changed for 4 h, and the liquid consumption in the bottle was recorded. The formula for calculating sucrose preference was: preference = (sucrose intake/total intake) × 100%.

**Surgery and drug treatment**
Mice were anesthetized with sodium pentobarbital (1%, 50 mg/kg) by intraperitoneal injection and placed in the stereotaxic apparatus (RWD, Shenzhen, China). Then, stainless steel guiding cannulas with stylets were bilaterally implanted into the hippocampus (AP: −2.0 mm; ML: ±1.5 mm; DV: −1.8 mm), and mice were allowed to recover for 7 days. Before defeat stress exposure, animals received bilateral intra-hippocampal infusions of A839977 (1 μg/0.5 μl/side). We used aCSF (Artificial Cerebrospinal Fluid) as control at a continuous rate of 0.1 μl/min via a micro-infusion pump (Harvard Apparatus, Shanghai, China). During all experiments, injector needles remained in place for 5 min before being pulled out.

**Brain slices preparation**
Mice were anesthetized with 1.5% isoflurane and transcardially perfused with heparinized saline as well as 4% paraformaldehyde. Brain extraction after euthanasia of cervical dislocation in mice. The brain of mice was extracted and stored in gradient sucrose solution for dehydration. After OCT (SAKURA, Japan) embedding, the 35-μm-thick frozen brain sections were pasted.

**Ca²⁺ signaling analysis**

**$Ca^{2+}$ signaling analysis**
BV2 or primary microglia were incubated in 24-well plates. Drugs were added to the bath medium. After adding 5 μM Fluo-4 AM (Thermo Fisher, F-14201), Rhod-2 AM (YEASEN, Shanghai, China) or Mag-Fluo-4 AM (AAT Bioquest, California, USA) to HBSS (Hank's Balanced Salt Solution, Corning, New York, USA), the cells were incubated at 5% $CO_2$ and 37 °C in dark environment for 30–60 min. Then they were washed twice using HBSS and incubated again with HBSS at room temperature for 30 min. Before analysis, cells were kept in HBSS.

Fluorescence images were obtained using an inverted fluorescence microscope (CKX53, OLYMPUS, Shinjuku, Japan). The $E_x/E_m$ of Fluo-4 AM was 494 nm/516 nm; $E_x/E_m$ of Rhod-2 AM was 549/578 nm; and $E_x/E_m$ of Mag-Fluo-4 AM was 494/516 nm. Quantification of the fluorescence was performed using ImageJ.

**Western blot analysis**
Cells or hippocampal tissues (Mice were anesthetized with sodium pentobarbital by intraperitoneal injection and brain extraction through cervical dislocation euthanasia method) were collected in a protein lysate buffer containing RIPA (Millipore, Massachusetts, USA) with 1% PMSF (Sigma, Missouri, USA) and 1% protease inhibitor cocktail (MCE, State of New Jersey, USA). BCA Standard Solution (Sigma, Missouri, USA) was used to calculate protein concentration. The denatured protein samples were analyzed using SDS-PAGE gel and transferred to a polyvinylidene difluoride (PVDF) membrane. After blocked the membranes and then incubated in the primary antibody at 4 °C overnight. After washing with Tris Buffered Saline with Tween (TBST) three times, the membranes were incubated with HRP-conjugated secondary antibody for 1 h at room temperature. Immunodetection was performed using the Immun-Star™ Western C™ HRP chemiluminescence kit and Molecular Imager® ChemiDoc™ XRS system (Millipore, Massachusetts, USA). All uncropped Western blots are available in Supplementary information.

**Co-Immunoprecipitation (Co-IP)**
Adherent BV2 cells were collected into 1.5 ml EP tube, where binding buffer (50 mM Tris, 150 mM NaCl, 0.1% TritonX-100) and protease inhibitor cocktail (MCE, State of New Jersey, USA) were added. Meanwhile, binding buffer was used to wash Protein A/G magnetic beads. Then, 1.5 mg protein solution, 10 μl specific antibody, and 30 μl of protein A/G magnetic beads were mixed and left overnight at 4 °C. After magnetic separation, the magnetic beads were washed with binding buffer. Then, LDS Sample Buffer 4× was added to the beads and boiled at 95 °C for 10 min. Finally, the supernatant was collected by magnetic separation for Western blot analysis. All uncropped Western blots are available in Supplementary information.

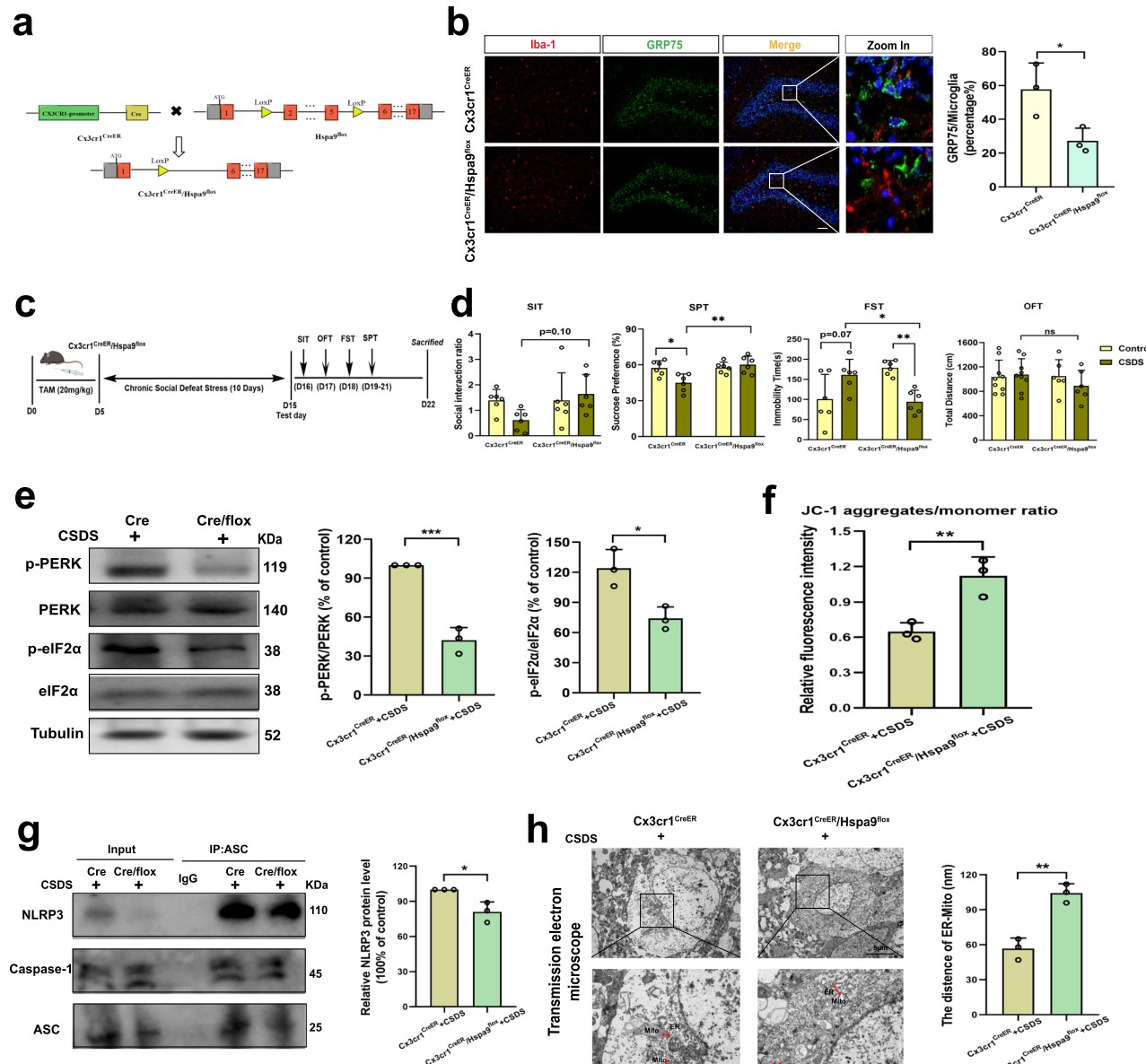

**Fig. 4 | Conditional knockdown of GRP75 in hippocampal microglia attenuates CSDS-induced depression-like behaviors. a** Traditional Cre-loxP method for generating GRP75 (*Hspa9*) gene deficient mouse models in microglia. Breeding scheme of the Cx3cr1[CreER/+] mice crossed with Hspa9[f/+]. **b** Representative immunofluorescent images (left panel) and percentage of positive microglia (right panel, n = 3; P = 0.0378) of GRP75 in Iba-1 cells in hippocampal DG from Cx3cr1[CreER/+] and Cx3cr1[CreER/+]/Hspa9[f/+] mice. GRP75 (green), Iba-1 (red), DAPI (blue). Scale bar: 20 μm. **c** Experimental timeline of tamoxifen (TAM) injection, CSDS, and behavioral testing. Figure 4c was created with BioRender.com released under a Creative Commons Attribution-Non Commercial-No Derivs 4.0 International license. **d** Behavioral tests including Social Interaction Test (SIT, Interaction, $F_{(1, 20)}$ = 2.986, P = 0.0994), sucrose preference test (SPT, Interaction, $F_{(1, 20)}$ = 8.064, P = 0.0101), forced swim test (FST, Interaction, $F_{(1, 20)}$ = 19.58, P = 0.0003), and open-field test (OFT,

Interaction, $F_{(1, 20)}$ = 0.9916, P = 0.3285) were performed (n = 6 in each group). **e** Western blot analysis of PERK and elf2α phosphorylation proteins from Cx3cr1[CreER/+] mice or Cx3cr1[CreER/+]/Hspa9[f/+] mice treated with CSDS (n = 3 mouse tissue samples; P = 0.0005; P = 0.0167). **f** Ratios of JC-1 aggregates to JC-1 monomer in two groups (n = 3 mouse tissue samples; P = 0.0099). **g** Representative blots of CO-IP (left panel) and quantification of the protein changes of NLRP3 in two groups (n = 3 mouse tissue samples; P = 0.0183). **h** Ultrastructural analysis of ER-mitochondria contact in two groups. Representative TEM images of ER-mitochondria contact (left panel) and the quantitative analysis of the distance between mitochondria and the ER (right panel, n = 3 mitochondria in 3 fields per condition; P = 0.0024). Scale bar: 5 μm. Student's *t* test or Two-way ANOVA with Tukey's multiple comparisons test. Data are expressed as mean ± SEM. ns, *p* > 0.05, *\*p* < 0.05, *\*\*p* < 0.01, *\*\*\*p* < 0.001. Source data are provided as a Source Data file.

## RNA isolation and qPCR

The RNA of mouse hippocampus tissue (Mice were anesthetized by intraperitoneal injection of pentobarbital sodium and euthanized by cervical dislocation) was isolated using TriZol reagent (Takara, Japan). The 260/230 values of all RNA samples were ≥ 1.8. Reverse transcription was performed using PrimeScript (Takara, Japan). qPCR using SYBR green (Takara, Japan) was carried out with an Applied Biosystems 7900HT RT−PCR system with the following cycle parameters: 2 min at 95 °C; 40 cycles at 95 °C for 15 s, 59 °C for 30 s, 72 °C for 33 s; and

graded heating to 95 °C to generate dissociation curves for confirmation of single-PCR products. Data were analysed by comparing C(t) values of conditions tested using the ΔΔC(t) method. Related primers are listed in the **S.** Table S1.

## Proximity Ligation Assays (PLA)

Before PLA[54], Cells were cultured on coverslips coated with Poly-L-Lysine (PLL) and fixed with 4% paraformaldehyde, and then immersed in 0.1% Triton X-100 in PBS for 10 min. Anesthesia of mice, method of

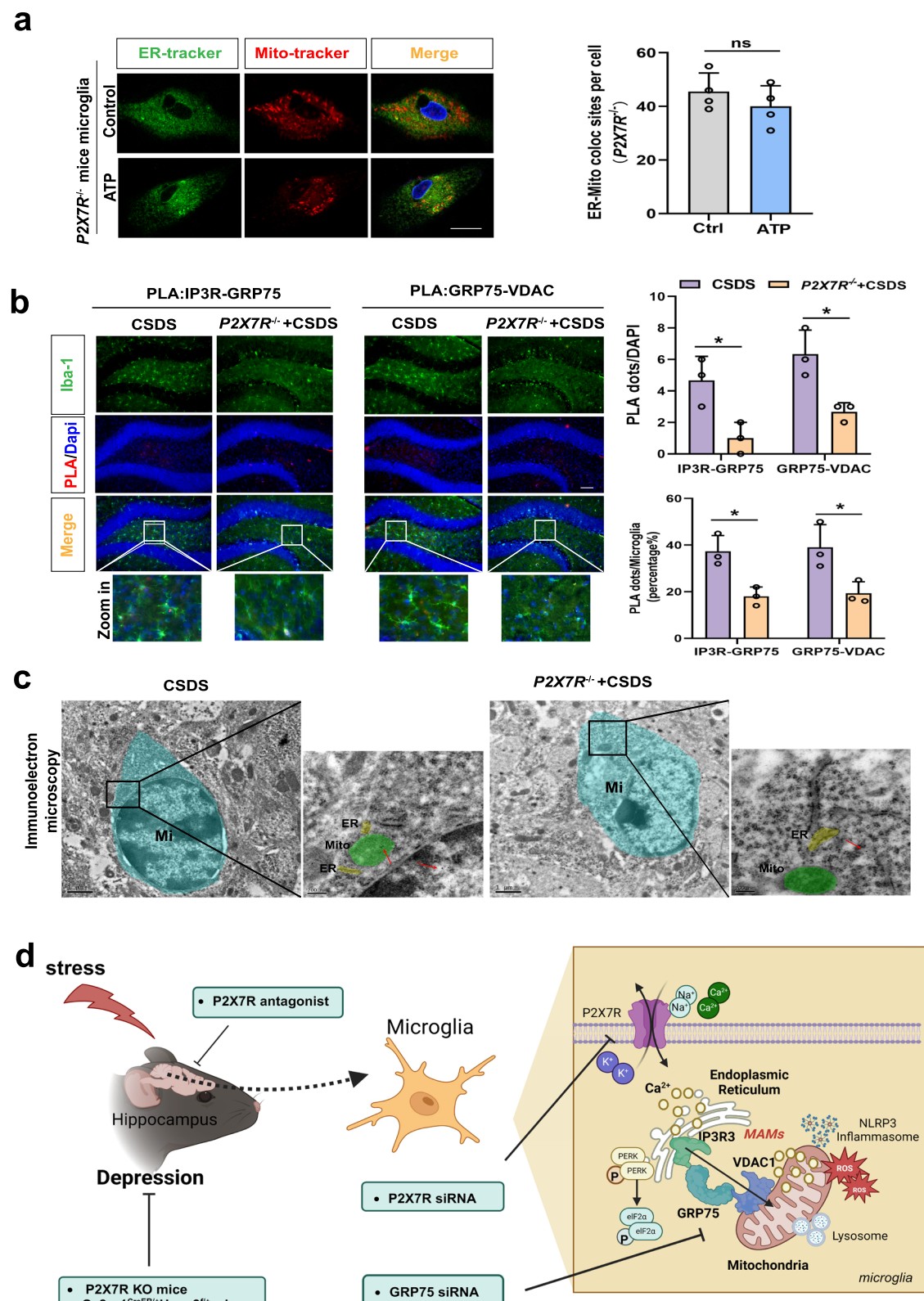

execution, and brain perfusion frozen sections were prepared as before.

Cell climbing patch or brain patch blocking was done using the Duolink blocking solution (Duolink® PLA Kit, Sigma-Aldrich, Missouri, USA) for 1 h at 37 °C. After the blocking solution was absorbed, diluted primary antibody solution (IP3R-3, GRP75 and VDAC1, 1:200) was added and the slides were kept overnight at 4 °C. Then, the slides were rinsed with Wash Buffer A. Subsequently, PLUS and MINUS secondary PLA probes were diluted in Duolink ® Antibody Diluent and incubated for 1 h at 37 °C. After washing with Buffer A twice, and dripping in the ligase solution, the probes were incubated at 37 °C for 30 min. After the probes were washed with Buffer A, an amplification buffer containing polymerase was added, and the probes were kept at 37 °C in the dark for 100 min. Subsequently, the slides were washed twice for

**Fig. 5 | P2X7 Receptors mediate CSDS-induced alterations in MAMs within hippocampal microglia. a** Representative confocal images of primary microglia untreated (Control) or treated with ATP (1 mM) from *P2X7R⁻/⁻* mice stained for nuclei (Hoechst 33342, blue), endoplasmic reticulum (ER-Tracker, green), and mitochondria (Mito-Tracker, red, left panel) and quantification of the colocalization between ER and mitochondria (right panel). Scale bar: 20 µm; Student's *t* test (P = 0.3313). Data are expressed as mean ± SEM (n = 4 batches of mouse extracted cell samples). ns, *p* > 0.05. **b** Representative images of PLA targeting IP3R3-GRP75 or GRP75-VDAC1 interactions in the hippocampal DG (left panel) and Quantification of the PLA red fluorescent dots (Top line, Interaction, $F_{(1, 8)}$ = 0.000, P > 0.9999) and the percentage of PLA dots in microglia (Bottom line, Interaction, $F_{(1, 8)}$ = 0.001815, P = 0.9671) were performed using Image J (right panel). Scale bar: 20 µm; Microglia (Iba-1, green), Nuclei (DAPI, blue). Two-way ANOVA with Sidak's multiple comparisons test. Data are expressed as mean ± SEM (n = 3 mouse brain slices). *\*p* < 0.05. **c** Immunoelectron microscope images with higher magnification of the inset showing the intimate proximity between mitochondria (M) and ER in a gold labeled microglia (Mi) (Red arrow: dense black dots) located in the hippocampal DG from CSDS and *P2X7R⁻/⁻* mice with CSDS. Source data are provided as a Source Data file. **d** Schematic diagram of the involvement of mitochondria-associated membranes (MAMs) in mediating extracellular ATP (eATP)/P2X7R-induced microglial responses and chronic social defeat stress (CSDS)-induced depression-like behaviors. CSDS induces ER stress, structural and functional alterations in MAMs, and mitochondrial impairment in hippocampal microglia. Reduced GRP75 expression in microglia of Cx3cr1^CreER/+Hspa9^f/+ mice or knockout P2X7 in P2X7R⁻/⁻ mice results in decreased depressive behaviors, reduced NLRP3 inflammasome aggregation, and fewer ER-mitochondria contacts in hippocampal microglia during CSDS. Figure 5d was created with BioRender.com released under a Creative Commons Attribution-Non-Commercial-No Derivs 4.0 International license.

10 min each time with Wash Buffer B. Nuclei were stained in situ with DAPI culture medium. After sealing, the slides were examined under a confocal microscope (SP5; Leica, Wetzlar, Germany). Quantification of the PLA red fluorescent dots was performed using ImageJ.

### Lyso-Tracker red test
The lysosomes were detected using Lyso-Tracker Red (Beyotime, Shanghai, China). After being cultured on sterile culture dishes, each group of treated BV2 cells and primary cultured microglia were incubated with 50 nM Lyso-Tracker Red at 37 °C for 15–30 min. The cells were then washed with PBS to remove any excess lysosomal marker and visualized using an SP5 confocal microscope (SP5; Leica, Wetzlar, Germany).

### Measurement of mitochondrial ROS
Mitochondrial superoxide was detected using MitoSOX Red (Thermo Fisher Scientific, USA), a mitochondrial superoxide indicator. Cells cultured on dishes, after drug treated 5 µM MitoSOX Red and 50 nM Mito Tracker Green added and incubated (Beyotime, Shanghai, China) for 15 min at 37 °C. Intracellular ROS measured was using 2′,7′-dichlorofluorescein diacetate (DCFH-DA, Sigma-Aldrich), a cell-permeable ROS indicator. Cells were incubated with 10 µM DCFH-DA for 20 min at 37 °C. Fluorescence images were obtained using an inverted fluorescence microscope (CKX53, OLYMPUS, Shinjuku, Japan). The $E_X/E_m$ of MitoSOX Red was 510 nm/580 nm and $E_X/E_m$ of DCFH-DA was 488/525 nm.

### Imaging of ER-mitochondria colocalization
Primary microglia were grown in glass-bottom dishes (35 mm, WHB, Shanghai, China). Cells were washed twice with PBS and then incubated with fresh culture medium containing 20 nM Mito-Tracker Red (Thermo Fisher Scientific, Massachusetts, USA) and 100 nM ER-Tracker Green (Thermo Fisher Scientific, USA) for 20–30 min at 37 °C, as well as 5 µM Hoechst, 33342 (Thermo Fisher Scientific, Massachusetts, USA) for 10 min. Then, the cells were washed with Live Cell Imaging Solution (Thermo Fisher Scientific, Massachusetts, USA). Imaging was performed using 63x oil objective (SP5; Leica, Wetzlar, Germany) and utilizing the Leica LAS software and ImageJ to observe colocalization.

### Mitochondrial transmembrane potential (ΔΨm) assay
Fluorescent probe JC-1 (BD Pharmingen, New York, USA) was used in the detection of mitochondrial membrane potential (MMP). After cells were pretreated with TG or ATP, JC-1 working solution was added to the culture medium and incubated at 37 °C for 30 min. After washing twice with cold dyeing buffer and the subsequent centrifugation, the cells were resuspended in the dyeing buffer. Quantification by flow cytometry (BD, FACSVerse, New York, USA) was used to detect mitochondria (EX = 488 nm), with red JC-1 aggregates in the FL2 channel and green JC-1 monomers in the FL1 channel (FL1, Em = 525 ± 20 nm,

FL2, Em = 585 ± 20 nm). The data were analyzed using Flow Jo Software (Treestar, FlowJo 10.6.2).

Mitochondria were extracted from the hippocampus (Mice were anesthetized by intraperitoneal injection of pentobarbital sodium and euthanized by cervical dislocation) using the Mitochondrial Separation Kit (Beyotime, Shanghai, China) according to the manufacturer's instructions, contains complete mitochondrial separation reagents, lysates, and storage solutions, with the vast majority of isolated mitochondria possessing mitochondrial physiological functions[55]. The extracted mitochondria were added to 96 well plate with black background and JC-1 dye (5 mg/ml) in a dark environment and cultured at 37 °C for 30 min. Molecular devices (SpectraMax ® Paradigm ®, California, USA) were used to measure the excitation (560 nm)/emission (595 nm) red fluorescence and excitation/emission (485 nm/535 nm) green fluorescence of each sample. Changes in mitochondrial membrane potential were expressed as the red/green fluorescence intensity ratio.

### Immunohistochemistry
Anesthesia of mice, method of execution, and brain perfusion frozen sections were prepared as before. The sections were first incubated with fluorescent blocking solution (Beyotime, Shanghai, China) for 1.5 h, and then incubated in a diluent containing Iba-1 antibody (#17198, 1:500, Cell Signaling technology, Danvers, MA, USA; ab289874, 1:100, Abcam, Cambridge Science Park, UK), Caspase-12 antibody (YT0654, 1:100, ImmunoWay, California, USA), VDAC1 antibody (Cat No. 55259-1-AP, 1:200, Proteintech, Chian), GRP75 antibody (sc-133137, 1:100, Santa Cruz Biotechnology, Montana, USA) or CD68 antibody (ab125212, 1:500, Abcam, Cambridge Science Park, UK) overnight at 4 °C. The samples were washed with PBS three times for 15 min each time, and then a secondary antibody (Alexa 488, 1:500, Abcam, Cambridge, UK) and DAPI Fluoromount-G (SouthernBiotech, Birmingham, UK) was used. Finally, the sections were observed using 20 × magnification lens under a microscope (DMi8; Leica, Wetzlar, Germany).

### Sholl analysis
Analyze according to previously reported methods[56]. Hippocampal IBA-1 immunoreactive microglia were imaged by using laser scanning confocal microscopy. Obtain a projected z-stack image of an orthogonal view with a thickness of 1 µm. Using Fiji to segment the z-stack image and store it as an 8-bit image. Use Fiji plug-in AnalyzeSkeleton(2D/3D) Analysis (http://imagej.net/AnalyzeSkeleton, version 3.6.8) for image analysis.

### Transmission and Immunoelectron microscopy analysis
First, cells were scraped and centrifuged (1500 rpm, 5 min). After discarding the supernatant, the cells were washed in PBS twice, centrifuged and agglomerated, while 2.5% glutaraldehyde was added to the supernatant and kept at 4 °C overnight. The cells were then postfixed in

1% osmium tetroxide in 0.1 M cacodylate buffer, dehydrated in graded ethanol and embedded in resin. The mice were perfused with 20 ml 0.9% NaCl (37 °C), 20 ml stationary liquid (37 °C, 0.25% glutaraldehyde) and 100 ml 0.25% glutaraldehyde (4 °C) by heart perfusion. Mice were anesthetized with sodium pentobarbital by intraperitoneal injection. Brain extraction through cervical dislocation euthanasia method and kept in 0.25% glutaraldehyde at 4 °C overnight. The 100-µm-thick sections of hippocampus were cut in sodium phosphate buffer (50 mM, pH 7.4) using a Leica VT1000S vibratome (Leica Biosystems) and stored in 1% osmium tetroxide for 30 min, followed by dehydration in ethanol and embedding in resin. The ultrathin sections of ~50 nm were achieved by using Leica EM UC7 Ultramicrotome (Leica, Wetzlar, Germany). These were collected in a nickel net.

For immunoelectron microscopy, sections were washed in 0.1% sodium borohydride (NaBH4) three times, 10 min each time, blocked for 1 h in 1% fetal bovine serum, and incubated with a Rabbit anti-Iba1/AIF-1 antibody (1:100, Cell Signaling Technology, Danvers, MA, USA) at room temperature for 2 h. Next, sections were incubated with secondary antibody conjugated to gold (5 nm colloidal gold, anti-rabbit IgG, 1:50, Sigma-Aldrich) at room temperature for 2 h and washed ten times with PBS for 10 min each time. For ultrastructure analysis of organelles, sections were stained with 2% uranyl acetate and observed using a Zeiss EM900 transmission electron microscope (Zeiss, Oberkochen, Germany).

### Bulk RNA sequence

**Dissociation of hippocampal tissue.** Mice were anesthetized with sodium pentobarbital by intraperitoneal injection. Brain extraction through cervical dislocation euthanasia method. Ten hippocampal tissues from five mice were extracted and combined into one experimental sample, and placed in 1.5 ml EP tubes with 1 ml cold 1×DPBS (Gibco™, USA). Enzymatic dissociation was performed using Adult Brain Dissociation Kit (130–107–677, Miltenyi Biotec, Cologne, Germany) according to the manufacturer's instructions. The tissue pieces (per up to 300 mg tissue) were transferred into a C Tube containing 1950 µl of enzyme mix 1 (Enzyme P and Buffer Z) and 30 µl of enzyme mix 2 (Enzyme A and Buffer Y). The probes were placed upside down in the sleeve of the gentleMACS Octo Dissociator with Heaters (130–096–427, Miltenyi Biotec), and subjected to the gentleMACS Program (37 °C _ABDK_01) for about 30 min.

**MACS magnetic beads sorting of microglia.** The supernatant was discarded after the dissociated solution was filtered through a 70 µm strainer (130–098–462, Miltenyi Biotec) and centrifuged at low speed (400 × *g*, 10 min, 4 °C). Precipitates were incubated for 15 min at 4 °C with Anti-CD11b Magnetic Microbeads (130–093-634, Miltenyi Biotec). 1 ml PB buffer was added to dilute and the probes were pretreated in an LS column (130–042-401, Miltenyi biotec) with PB buffer. The column was fixed in a magnetic separator (130–042-302, Miltenyi Biotec) and cell suspension was applied into the column. The column was washed with 2 ml PB buffer to flush out the magnetically labeled cells into a 1.5 ml EP tube. The magnetically labeled cells were centrifuged at 400 × *g* for 10 min. The supernatant was discarded, while the precipitate was suspended in 1 ml TRIzOl Reagent (Invitrogen, Carlsbad, CA, USA).

**RNA extraction.** Total RNA was extracted from the tissue using TRIzol® Reagent according the manufacturer's instructions. Then RNA quality was determined by 5300 Bioanalyser (Agilent) and quantified using the ND-2000 (NanoDrop Technologies). Only high-quality RNA sample (OD260/280 = 1.8–2.2, OD260/230 ≥ 2.0, RIN ≥ 6.5, 28 S:18 S ≥ 1.0, >1 µg) was used to construct sequencing library.

**Library preparation and sequencing.** RNA purification, reverse transcription, library construction, and sequencing were performed at Shanghai Majorbio Bio-pharm Biotechnology Co., Ltd. (Shanghai, China) according to the manufacturer's instructions (Illumina, San Diego, CA). The microglia RNA-seq transcriptome library was prepared following Illumina® Stranded mRNA Prep, Ligation from Illumina (San Diego, CA) using 1 µg of total RNA. Shortly, messenger RNA was isolated according to polyA selection method by oligo(dT) beads and then fragmented by fragmentation buffer firstly. Secondly double-stranded cDNA was synthesized using a SuperScript double-stranded cDNA synthesis kit (Invitrogen, CA) with random hexamer primers (Illumina). Then the synthesized cDNA was subjected to end-repair, phosphorylation, and 'A' base addition according to Illumina's library construction protocol. Libraries were size selected for cDNA target fragments of 300 bp on 2% Low Range Ultra Agarose followed by PCR amplified using Phusion DNA polymerase (NEB) for 15 PCR cycles. After quantified by Qubit 4.0, paired-end RNA-seq sequencing library was sequenced with the NovaSeq 6000 sequencer (2 × 150 bp read length).

**Quality control and read mapping.** The raw paired end reads were trimmed and quality controlled by fastp[57] with default parameters. Then clean reads were separately aligned to reference genome with orientation mode using HISAT2[58] software. The mapped reads of each sample were assembled by StringTie[59] in a reference-based approach.

**Differential expression analysis and Functional enrichment.** To identify DEGs (differential expression genes) between two different samples, the expression level of each transcript was calculated according to the transcripts per million reads (TPM) method. RSEM[60] was used to quantify gene abundances. Essentially, differential expression analysis was performed using the DESeq2[61] or DEGseq[62]. DEGs with $|log2FC| \geq 1$ and FDR ≤ 0.05(DESeq2) or FDR ≤ 0.001(DEGseq) were considered to be significantly different expressed genes. In addition, functional-enrichment analysis including GO and KEGG were performed to identify which DEGs were significantly enriched in GO terms and metabolic pathways at Bonferroni-corrected P ≤ 0.05 compared with the whole-transcriptome background. GO functional enrichment and KEGG pathway analysis were carried out by Goatools and KOBAS[63], respectively.

**Alternative splice events identification.** All the alternative splice events that occurred in our sample were identified by using recently releases program rMATS[64]. Only the isoforms that were similar to the reference or comprised novel splice junctions were considered, and the splicing differences were detected as exon inclusion, exclusion, alternative 5′, 3′, and intron retention events.

### Statistical analysis

Statistical analyses were performed with GraphPad Prism 8.0 (GraphPad Software, San Diego, CA). Unless otherwise stated, data were presented as mean ± SEM (standard error of mean). We performed unpaired Student's *t* test, one-way ANOVA or two-way ANOVA for assessing the significance of differences between different groups, as indicated in the figure legends. $P < 0.05$ was taken as indicator of statistical significance.

### Reporting summary

Further information on research design is available in the Nature Portfolio Reporting Summary linked to this article.

## Data availability

Bulk RNA sequencing data that support the findings of this study has been deposited in NCBI BioProject with the primary accession code PRJNA1119092 and in SRA with the primary accession codes SRX24775092, SRX24775093, SRX24775094, SRX24775095, SRX24775096, and SRX24775097 (https://www.ncbi.nlm.nih.gov/sra/PRJNA1119092). Source data are provided as a Source Data file. All other data is available from the corresponding author upon request. Source data are provided with this paper.

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

## Acknowledgements

We acknowledge Adam Pilot for commenting on grammar, spelling, or format, and the technical assistance of Wen -Wen Zhang for helping with PLA experiments. We appreciate BioRender.com for creating the Experimental flow chart and Graphical Abstract. This work was supported by grants from National Natural Science Foundation of China (82174499, 31930042 and 81671349), National Key R&D Program of China (2017YFB0403803), Shanghai Municipal Science and Technology Major Project (No.2018SHZDZX01), Development Project of Shanghai Peak Disciplines-Integrated Chinese and Western Medicine, Building Project for Innovative Team of National Traditional Chinese Medicine, Innovative Research Team of High-Level Local Universities in Shanghai and ZJ Lab.

## Author contributions

J.Y. and Y.Q.Z. performed study concept and design. J.R.Z. and S.Y.S. performed most of the experiments. M.Y.Z., W.L., and L.F.L. conducted the behavioral tests. J.R.Z., S.Y.Y., and Z.Q.S. performed the molecular biology experiments. Z.Q.S., Q.Q.H., and B.L. undertook the statistical analysis, J.Y. and J.R.Z. wrote the manuscript. All authors read and approved the final paper

## Competing interests

The authors declare no competing interests.

## Additional information

Jia-Rui Zhang[1,5], Shi-Yu Shen[1,2,5], Meng-Ying Zhai[1], Zu-Qi Shen[1], Wei Li[1], Ling-Feng Liang[1], Shu-Yuan Yin[1], Qiu-Qin Han[1], Bing Li[3], Yu-Qiu Zhang ®[2] & Jin Yu[1,4] ✉

[1]Department of Integrative Medicine and Neurobiology, School of Basic Medical Sciences, Shanghai Medical College, Fudan University, Shanghai 200032, China. [2]Department of Translational Neuroscience, Jing'an District Centre Hospital of Shanghai, State Key Laboratory of Medical Neurobiology and MOE

Frontiers Center for Brain Science, Institutes of Brain Science, Fudan University, Shanghai 200032, China. [3]Center Laboratories, Jinshan Hospital of Fudan University, Shanghai 201508, China. [4]Shanghai Key Laboratory of Acupuncture Mechanism and Acupoint Function, Fudan University, Shanghai 200433, China. [5]These authors contributed equally: Jia-Rui Zhang, Shi-Yu Shen. ✉e-mail: yujin@shmu.edu.cn

