## [Peer Review File · Nature Communications]

Augmented Microglial Endoplasmic Reticulum-Mitochondria Contacts Mediate Depression-Like Behavior in Mice Induced by Chronic Social Defeat StressReviewers' Comments:

Reviewer #1 (Remarks to the Author):

In the article "Chronic enrichment of microglial endoplasmic reticulum- 1 mitochondria contact leads to calcium dependent mitochondrial 2 dysfunction in depression" by Zhang et al. the authors investigated the role of P2RX7 in depressive-like behaviors in the social-defeat stress mouse model, and the underlying mechanisms. With a series of mice and in vitro experiments they show that microglia P2XR7 is involved in social-defeat stress-induced depressive-like behavior with a role for ER stress, leading to inflammasome activation. They show the location of these processes to take place in the mitochondria-associated membranes. The topic of the paper is of interest to the field and the paper consists a lot of interesting data. Major concerns are related to the models that are the foundation of this paper.

Major comments:

1. A large part of the in vitro experiments have been performed with the BV-2 cell line. The validity of the BV-2 cell line (and all other immortalized microglia cell lines) as a model for microglia is under debate (PMID 24316888, 29775624). Therefore, the choice for using BV-2 cells should have been justified, the limitations clearly stated and more (validation) experiments should have been performed with primary microglia and these should be shown in the main figures.
2. Details and QCs on the mouse lines used are lacking. The catalogue number of Jackson that was mentioned in the paper refers to the NOD.129P2(B6)-P2rx7t1Gab/DvsJ mouse and that a NOD control should be used. Instead, the authors used C57BL/6J mice of another company. Validation of the KO of P2RX7 should be shown.
3. It is unclear why the authors use the long social defeat model in some experiments, and why they switch to sub-SDS in others.
4. More details for the RNA-seq analysis are needed: QC steps, filter steps, statistics behind differential gene expression analysis. Also, a validation on the protein level using immunostainings would make the changes more convincing.

Minor comments:

1. The manuscript is sometimes hard to understand and could use some language editing. As an example: "High levels of external stimuli are associated with the development of depression". It is unclear what the authors mean by high levels of external stimuli. Stress? Inflammatory triggers? Cell damage products?
2. " Among these, pro inflammatory genes, 120 including Ccl8, Cxcl2 and Il11b, as well as CD86, the primary 'marker' for M1 microglia". Please update following recent discussion on the M1/M2 terminology (PMID 27459405)
3. There is a lot more literature on P2RX7 in depression (reviewed in PMID 31174279). For instance, the genetic associations between depression and P2RX7, which would be interesting to refer to in the paper.

Reviewer #2 (Remarks to the Author):

Review of Chronic enrichment of microglial ER-mito contact leads to calcium dependent mitochondrial dysfunction in depression

This is a paper by Zhang et al. that attempts to show that a chronic enrichment of microglial ER-Mitochondrial contact leads to a calcium-dependent microglial dysfunction in depression. The authors nicely combine in vivo and in vitro approaches to investigate microglial subcellular structure dysfunction and its contributions to stress-induced depression-like behaviors. While the central story (which is sometimes difficult to appreciate) is quite strong, there are several major and minor concerns discussed below.

Major Concerns

[1] The major concern from this reviewer is that the study does not provide sufficient support to

substantiate the claim of hippocampal microglial specific function. BV2 cells where most of the in vitro studies are conducted are not necessarily hippocampal microglia. P2X7 which is the initial recipient of the ATP signal is not selectively localized to hippocampal microglia. Importantly, they find changes in hippocampal microglia but they do not check to see that the effect does not occur in non-hippocampal microglia. Furthermore, while hippocampal microglial activation (by both Iba1 fluorescence and RNAseq) in the CSDS were the focus, it is not clear that the microglial phenotypes observed do not occur in non-hippocampal regions. Such controls are lacking.

Further lacking are details for the important results from Fig 8a-c on the localization of ER stress inhibitor / mitochondrial protector treatment. If the treatment was broad (rather than specifically targeting the hippocampus), it would be a great finding but would not support the hippocampal microglia specific claim.

In addition, even if these drugs were specifically delivered to the hippocampus, since ER and mitochondria (and the corresponding CSDS-induced stress) are not selectively localized to microglia, the lack of an assessment of consequences of these drug treatments on non-microglial cells warrants caution that the effects occur through microglia (even in the hippocampus). At the least, the authors should acknowledge and discuss this caveat. Better still would be studies to examine and indeed rule out e.g. beneficial effects of these drugs on astrocytic or neuronal ER / mitochondrial health which could explain the findings independent of microglia. In this vein, the claim in the title suggesting microglial ER-mito contact as causative is not substantiated.

This reviewer suspects that the authors want to argue that CSDS alters hippocampal circuits specifically (e.g. increased ATP release here) which in turn activates microglia and leads to everything discussed. This is then the reason why hippocampal microglia are the focus. If this is the case, this needs to be more explicitly and clearly mentioned because as it currently reads like the study wants to claim something specifically unique about hippocampal microglia without much support [see also point #4 below]

[2] A second concern with the manuscript is the poor images. All the IHC images are problematic because the cells can barely be seen. For example, in Fig. 1d, while there is roughly a 50% increase in Iba1 intensity according to the quantification, it is not clear that this is the case with the corresponding images. This is true for Fig. 2d and 7c as well. Moreover, in these cases intensity of Iba1 is measured to be increased but it is not clear if this is because of an increased number of cells (i.e. density) or expression of Iba1. For Fig. 7c, the increase is referred as a density increase (line 287). This can/should be confirmed by ki67 staining as a proliferation marker. Especially for Fig. 2d, differences are not evident. Are different methods used in the analyses? No details on the Iba1 analysis is provided in the methods. This reviewer recommends that the authors present images with greater magnification so that the cells can be better visualized. Moreover, it is not clear the usefulness of including the DAPI stain / merge image to indicate that the DG is examined. The manuscript is not focused on the DG but on hippocampus so given more space for visualization would be preferred that providing the DG context. Images like the zoomed in images of Fig. 6h are preferred, though the quality of those images can be improved.

In addition, in Fig. 2a images one can barely see anything. Fig. 3f is also barely visible because it is so small. Scale bars are barely visible in 4f. Moreover, green/red colors should be avoided to accommodate color blind readers. For the EM images e.g. in Fig. 6d, it is not clear that the red arrows are pointing to the same things in the control and CSDS conditions. Careful organization and presentation of the Figures would improve the manuscript.

[3] A third concern with the manuscript is the really complicated flow. This is a difficult manuscript to follow and needs to be read carefully (even multiple times) to be really understood. It feels that there are three parts to the paper. The first part (Fig. 1 & 2) seeks to establish microglial P2X7R signaling in the mediation of depressive like behaviors in a chronic social defeat stress model. First, it's not clear why this model was used over the chronic unpredictable mild stress model where P2X7R roles have been previously shown. Both models seem to generate depressive-like symptoms so why use a new model? The NLRP3 assembly features in the former model but doesn't seem to be relevant for the current model/story or even in the final model schematic. Why

mention it? Pro-inflammatory microglial genes are mentioned in Fig. 1 following CSDS but the rest of the parts study does not really examine pro-inflammatory gene expression. These all serve to be distractions from understanding the focus of the study since they are not crucial to the central point(s) of the study. Also, although P2X7R KO mice are available, they do not validate the antibody used on the KO mice to confirm the trustworthiness of that antibody in P2X7Rs. Could the authors do this?

Having confirmed microglial activation and P2X7R roles in the first part, the authors move to the second part to tease out intracellular mechanisms (Fig. 3-5) of ER-Mito dysfunction in ATP toxicity that presumably occurs in response to CSDS (though this is not shown directly or discussed). These studies are done in BV2 cells in vitro which have nothing specifically related to "hippocampal microglia". This provides some disconnect with the previous part that the authors should try to limit. Moreover, there are concerns that in vitro microglia (and especially BV2 cells) do not recapitulate in vivo microglial phenotypes that the authors don't address.

The final part is a return to the in vivo CSDS model to confirm ER-mito damage that was observed in vitro (Fig. 6). Then they confirm that ER stress is sufficient to elicit the depression-like behaviors without CSDS (Fig. 7). This is great but the treatment is not specific to microglia so there is a serious gap in this flow with the story which the authors don't address. Finally, the authors prevent the ER stress and mito dysfunction pharmacologically to improve ER-mito histology and depressive behaviors. But again this treatment is not specific to microglia. Given the commencement with P2X7R, it also feels like an incomplete circle to not test the drug in KO mice as well.

Therefore, to this reviewer, while the template of the story is nice: the combination of in vivo and in vitro approaches etc, the links made between each of these three parts are tenuous at times. Independently each part could stand but together the authors need to do a better job in making the appropriate connections and claims and at least discussing their study limitations. It is the view of this reviewer that simplifying the flow to be more focused will make more compelling manuscript. Some panels of these current main figures can be moved to Supplemental Figures to allow for more space to explain and visualize some of these critical (especially images) panels [see also point #2]

[4] A final major concern is the language of the manuscript. While much of the language is understandable, some of it makes for very difficult reading. For example, the Abstract was very difficult to understand for this reviewer. "proposed" in line 31 should be better replaced with something like "investigated". The results were simply stated [lines 31-41] and it was difficult to initially grasp whether the authors performed these experiments and generated these results, or these are background facts. Descriptors like "we find/found that...", "we show/showed that...", "we discover/discovered that...". In line 40, "especially" in microglia is mentioned but this is somewhat misleading because they only study microglia throughout the manuscript. The use of "while" on line 109 is quite confusing as it kind of sets off a possible contrast but the two ideas are not contrasting. The authors should incorporate a careful and thorough proof-reading / editing of the manuscript in a subsequent submission especially with the help of (an) English-proficient editor(s).

Minor Concerns

[1] Sample sizes used for in vitro experiments were unclear e.g. "n = 5 per condition" 5 cells? 5 experiments? Fields of view? How many cells per experiment/fields of view etc? For Fig. 3a-c; or "n = 4" for Fig. 3d 4 what? Cells? experiments? How many cells? etc. The authors should provide these types of information throughout the Figures.

[2] Fig. 2d should have data for the basal fluorescence without CSDS to confirm that P2RY12 does not regulate basal Iba1 expression.

Reviewer #3 (Remarks to the Author):

In their manuscript Zhang et al have attempted to highlight the role of ER-mitochondria tethering in mouse models of depression. The authors have also performed a wide array of experiments in cell lines and primary microglial cells to support their in vivo findings and have made an effort to provide a more sophisticated insight of the underlying mechanisms.

I regret to write in this review that this paper has many major issues on every level.

I think that this paper suffers from poor writing and in several cases poor interpretation and analysis of the data as well as technique application. Although I will give some examples to justify my opinion, I will not go through every detail. I sincerely hope that the authors find the following criticism constructive.

Firstly, in terms of writing, the paper lacks logical structure. Already the abstract, for someone who did not carefully read the entire manuscript, is very complex and disconnected. It is even hard to understand the main point of the paper when psychiatric disorders are presented with mitochondrial impairment as a culprit (reference 22) and the authors focus on microglial cells and not neurons. Used experimental approaches are not justified (e.g., why did the authors choose to perform an experiment in which they placed aCSF in mice? Or why was a subCSDS used? Why were A839977 and BzATP chosen?) and the presented points do not follow any obvious rationale (e.g., why did they look if Ca²⁺ is gated into mitochondria (lines 165-166)? How are the data presented in the 1st section of the results "further confirm the function of P2X7Rs" (line 105)?). The article needs to be completely restructured to connect its various aspects and make it easier to follow. The working hypothesis needs to become more clear and aligned with the data and all results should be part of a one bigger connected story. In the present form of the article, the findings seem unconnected and irrelevant to one another while the authors still jump into rather daring conclusions (lines 174-175, lines 221-222, lines 230-233 etc). It is quite clear that in an effort to connect so many different concepts (depressive-like phenotypes, microglia activation, inflammatory markers and complexes, P2X7R receptors, ER-mitochondria tethering and Ca²⁺ dynamics) the logical connections have been majorly neglected. Perhaps removing or condensing some of the results will make the points of the work sharper and in focus.

Despite the fact that major restructuring of the paper would comprise a significant improvement, I still think that in several cases the data analysis needs extensive implementations. For example, in many cases it is unclear how many independent biological replicates have been used (e.g, Figures 1d, 3a, 3b, 3c, 3f). Also, in some other instances the chosen statistical tests were parametric when only 3 points/group were compared and a non-parametric test would be more appropriate (e.g., figures 1d and 2d). Additionally, in some figures (e.g, 1d and 2d) the control bars do not have any error bars that is not only biologically impossible, but also statistically incorrect while at the same time it biases any applied statistical test to give significant results. Another sign that the authors should consult with an expert regarding their data analysis, is the Venn diagram in Figure 1f. The data on this diagram have no biological or statistical meaning based on what is described in the text. In conclusion, this paper lacks on many occasions proper statistical analysis. This is important because correct statistical analysis or addition of more data points might change the significance of the data, ergo their interpretation in this particular work.

Finally, I strongly disagree with some of the experimental approaches or selected experimental methods. The data presented in figures 2b and 2c are a good example to support my point.

Although it is not explained in the text (the lack of sufficient explanation of the data is spread throughout the manuscript), it seems that the aCSF+CSDS group is set as a control and based on its performance the authors have concluded that A839977+CSDS mice have impeded development of depressive-like phenotypes. However, if one compares the performance scores of this figure with the ones of figure 1c, it is clear that A839977+CSDS have the same score with CSDS animals for SIT, which is around 120. This clearly indicates that figure 2c lacks vital controls and the substitution of a proper control with aCSF+CSDS affects the interpretation of an experiment that is central for the first part of the study. Another main issue is the use of an ER stain and mitotracker to determine the proximity of ER and mitochondria, an assay that has been used extensively in the paper (e.g. Figures 4 and 5) and supports several major claims although the method is inappropriate. The 2 accepted methods to determine ER-mitochondria proximity are TEM and PLA. Curiously, the authors did perform these assays in some instances (Figure 4b and Figure 4f, respectively) but failed to apply them in others while at the same time they based many of their conclusions on the results given by an inappropriate method.

Overall, the authors should take this opportunity and consider the above comments. Improving the quality of their data as well as restructuring their story has the potential of bringing forward impactful science. However, I am afraid that, in its present form, this work is not suitable for publication. I wish I could deliver a more positive opinion and once again, I sincerely hope that the authors find this criticism helpful.

Dear Reviewers,

I would like to express our sincere gratitude for the invaluable feedback provided by each of you.

Undeterred by the setback, we took your feedback to heart and made significant efforts to enhance our study. Among the steps taken, we meticulously constructed a transgenic mouse model and conducted additional experiments in direct response to your insightful comments. Your feedback has guided our revisions, and we believe the changes made have significantly strengthened the quality of our work.

Enclosed with this letter are your comments and our detailed responses, illustrating our commitment to addressing each point raised during the review process. The comments are in *italics*, and our responses are in regular text, highlighted in **blue**. The edits in the manuscript are presented in regular text with **red** highlights.

We recognize the importance of maintaining high standards at *Nature Communications* and genuinely appreciate the thorough review process that ensures the quality of the published work. We sincerely hope for your understanding and kindly request a re-review of our revised manuscript.

Thank you for your time and thoughtful review. We eagerly anticipate the possibility of contributing our work to the esteemed pages of *Nature Communications*.

Best regards,

Jin Yu Ph.D

Professor in Department of Integrative Medicine and Neurobiology,

State Key Laboratory of Medical Neurobiology,

School of Basic Medical Sciences,

Institutes of Brain Science, Brain Science Collaborative Innovation Center,

Shanghai Medical College, Fudan University,

Shanghai, 200032, China

E-mail: yujin@shmu.edu.cn

Reviewer comments are in *italics*, and author responses are in regular text, highlighted in **blue**.

Reviewer #1:

In the article “Chronic enrichment of microglial endoplasmic reticulum- 1 mitochondria contact leads to calcium dependent mitochondrial 2 dysfunction in depression” by Zhang et al. the

authors investigated the role of P2RX7 in depressive-like behaviors in the social-defeat stress mouse model, and the underlying mechanisms. With a series of mice and in vitro experiments they show that microglia P2XR7 is involved in social-defeat stress-induced depressive-like behavior with a role for ER stress, leading to inflammasome activation. They show the location of these processes to take place in the mitochondria-associated membranes. The topic of the paper is of interest to the field and the paper consists a lot of interesting data. Major concerns are related to the models that are the foundation of this paper.

Major comments:

1. A large part of the in vitro experiments have been performed with the BV-2 cell line. The validity of the BV-2 cell line (and all other immortalized microglia cell lines) as a model for microglia is under debate (PMID 24316888, 29775624). Therefore, the choice for using BV-2 cells should have been justified, the limitations clearly stated and more (validation) experiments should have been performed with primary microglia and these should be shown in the main figures.

RESPONSE: Thank you for your suggestion. We have addressed the concern by conducting additional experiments with primary microglia, and the results are now included in the main figures, specifically presented in Fig. 2.

2. Details and QCs on the mouse lines used are lacking. The catalogue number of Jackson that was mentioned in the paper refers to the NOD.129P2(B6)-P2rx7t1Gab/DvsJ mouse and that a NOD control should be used. Instead, the authors used C57BL/6J mice of another company. Validation of the KO of P2RX7 should be shown.

RESPONSE: Please accept my sincerest apologies for any confusion caused by my mistake. The correct catalogue number for the P2X7 knockout (KO) mice we obtained from Jackson Lab is 005576^{1,2}. We have rectified this error in the appropriate methods section. It's worth noting that we procured this strain from The Jackson Laboratory over a decade ago and have consistently crossed them with C57BL/6J mice acquired from Slack Laboratory Animals Ltd (Shanghai, China) for well over 10 generations. As a result, we can reasonably assume that their genetic background closely aligns with that of C57BL/6J mice from Slack Laboratory Animals Ltd. Furthermore, we have conducted verification of the P2RX7 knockout, and the relevant data is included in Supplementary Figure 2A, which has been submitted together.

3. It is unclear why the authors use the long social defeat model in some experiments, and why they switch to sub-SDS in others.

RESPONSE: The choice between the chronic social defeat model and sub-SDS was made based on the specific objectives of our experiments. The chronic social defeat model is a well-established animal model for studying stress-induced depression and anxiety, providing insights into the involvement of microglia and the intracellular-related pathological abnormalities associated with these conditions. On the other hand, sub-SDS was employed to examine the impact of endoplasmic reticulum stress inducers on stress susceptibility in animals. In response to the feedback from another reviewer, the results from this section have been excluded in the revised version.

4. More details for the RNA-seq analysis are needed: QC steps, filter steps, statistics behind differential gene expression analysis. Also, a validation on the protein level using immunostainings would make the changes more convincing.

RESPONSE: Thank you for your valuable suggestion. We have provided additional details on several key steps of the RNA-seq analysis, encompassing RNA extraction, library preparation, sequencing, quality control, read mapping, differential expression analysis, and functional enrichment. Furthermore, to enhance the robustness of our findings, we conducted immunohistochemical staining on the endoplasmic reticulum stress marker Caspase-12 and the mitochondrial damage marker gene VDAC1, aligning with the sequencing results. These immunostainings are presented in Figure S1A-B for your reference.

Minor comments:

1. The manuscript is sometimes hard to understand and could use some language editing. As an example: "High levels of external stimuli are associated with the development of depression". It is unclear what the authors mean by high levels of external stimuli. Stress? Inflammatory triggers? Cell damage products?

RESPONSE: We appreciate your valuable advice regarding the clarity of our manuscript. To address this concern, we have collaborated with a language editor to refine and rephrase certain sections, including the specific phrase you mentioned: "High levels of external stimuli are associated with the development of depression." The revised language aims to provide greater clarity and precision in conveying our intended meaning.

2. "Among these, pro-inflammatory genes, 120 including Ccl8, Cxcl2 and Il11b, as well as CD86, the primary 'marker' for M1 microglia". Please update following recent discussion on the M1/M2 terminology (PMID 27459405)

RESPONSE: We sincerely appreciate your constructive comments. Based on the recent discussion on M1/M2 terminology (PMID 27459405), we conducted additional PCR experiments to detect a broader range of microglia-related markers. The updated findings are as follows: "RT-PCR analysis of hippocampal tissues (Fig. S1G) revealed an increased expression of several surface receptors considered as microglia-sensitive markers, including CD68, Cx3cr1, MHC II, and CD40. Notably, Arg1 expression did not show a significant change following CSDS."

3. There is a lot more literature on P2RX7 in depression (reviewed in PMID 31174279). For instance, the genetic associations between depression and P2RX7, which would be interesting to refer to in the paper.

RESPONSE: We appreciate your insightful comment. In response to your suggestion, we have incorporated additional discussion on the genetic associations between depression and P2RX7 in the revised manuscript. For further exploration, we have referenced the following relevant literatures:

Troubat, R. et al. Brain immune cells characterization in UCMS exposed P2X7 knock-out mouse. *Brain Behav Immun* 94, 159-174 (2021). <https://doi.org:10.1016/j.bbi.2021.02.012>

Harvey, M., Belleau, P. & Barden, N. Gene interactions in depression: pathways out of darkness. *Trends Genet* 23, 547-556 (2007). <https://doi.org:10.1016/j.tig.2007.08.011>

Liu, J. et al. Genome-wide Mendelian randomization identifies actionable novel drug targets for psychiatric disorders. *Neuropsychopharmacology* 48, 270-280 (2023). <https://doi.org:10.1038/s41386-022-01456-5>

McQuillin, A. et al. Case-control studies show that a non-conservative amino-acid change from a glutamine to arginine in the P2RX7 purinergic receptor protein is associated with both bipolar- and unipolar-affective disorders. *Mol Psychiatry* 14, 614-620 (2009). <https://doi.org:10.1038/mp.2008.6>

Reviewer #2

Review of Chronic enrichment of microglial ER-mito contact leads to calcium dependent mito dysfunction in depression

This is a paper by Zhang et al. at attempts to show that a chronic enrichment of microglial ER-Mitochondrial contact leads to a calcium-dependent microglial dysfunction in depression. The authors nicely combine in vivo and in vitro approaches to investigate microglial subcellular structure dysfunction and its contributions to stress-induced depression-like behaviors. While the central story (which is sometimes difficult to appreciate) is quite strong, there are several major and minor concerns discussed below.

Major Concerns

[1] The major concern from this reviewer is that the study does not provide sufficient support to substantiate the claim of hippocampal microglial specific function. BV2 cells where most of the in vitro studies are conducted are not necessarily hippocampal microglia. P2X7 which is the initial recipient of the ATP signal is not selectively localized to hippocampal microglia. Importantly, they find changes in hippocampal microglia but they do not check to see that the effect does not occur in non-hippocampal microglia. Furthermore, while hippocampal microglial activation (by both Iba1 florescence and RNAseq) in the CSDS were the focus, it is not clear that the microglial phenotypes observed do not occur in non-hippocampal regions. Such controls are lacking.

RESPONSE: As noted by the reviewer, BV2 cells do not necessarily represent hippocampal microglia. Therefore, we conducted almost all experiments performed on BV2 cells using primary hippocampal microglia. The results of these experiments have been incorporated into the main figures. While it is acknowledged that P2X7 is not selectively localized to hippocampal microglia, prior studies have indicated that P2X7R is predominantly expressed in microglia. Moreover, the hippocampal eATP-P2X7R signaling pathway in microglia has been implicated in stress-induced neuroinflammatory and depressive phenotypes ^{1,2,3}. We also observed that CSDS can induce more active microglia in the medial prefrontal cortex (mPFC) (data shown in sFig.1i), which is consistent with existing literature ⁴. Additionally, literature reports have highlighted that CSDS can lead to more active microglia in the mPFC, characterized by morphological changes such as increased branching ⁴.

Further lacking are details for the important results from Fig 8a-c on the localization of ER stress inhibitor / mitochondrial protector treatment. If the treatment was broad (rather than specifically targeting the hippocampus), it would be a great finding but would not support the hippocampal microglia specific claim.

RESPONSE: The reviewer's insightful comments and suggestions are greatly appreciated. In response, we have expanded the experiments involving ER stress inhibitors/mitochondrial protectors based on your recommendations. However, we acknowledge that while these findings are valuable, they do not sufficiently support the claim of hippocampal microglia specificity. Consequently, we have decided to remove this portion of the experimental results from the manuscript and have accordingly revised the text. Thank you for bringing this to our attention.

In addition, even if these drugs were specifically delivered to the hippocampus, since ER and mitochondria (and the corresponding CSDS-induced stress) are not selectively localized to microglia, the lack of an assessment of consequences of these drug treatments on non-microglial cells warrants caution that the effects occur through microglia (even in the hippocampus). At the least, the authors should acknowledge and discuss this caveat. Better still would be studies to examine and indeed rule out e.g. beneficial effects of these drugs on astrocytic or neuronal ER / mitochondrial health which could explain the findings independent of microglia. In this vein, the claim in the title suggesting microglial ER-mito contact as causative is not substantiated.

RESPONSE: We greatly appreciate your insightful observations regarding the limitations of our experiments. Indeed, ER stress inhibitors or mitochondrial protectors do not specifically target ER or mitochondria in microglia, which hinders the elucidation of their roles in ER stress and mitochondrial damage in microglia during CSDS-induced depressive phenotypes. Therefore, we have removed the relevant results from this version of the manuscript.

Furthermore, we have utilized Cx3cr1^{CreER/+}/Hspa9^{flox/+} mice, which were bred last year, to investigate the involvement of ER-MAMs-mitochondrial alterations in microglia during CSDS-induced depressive phenotypes and microglial activation. These findings have been incorporated into the current manuscript for further discussion and analysis. Thank you for bringing attention to these important considerations.

This reviewer suspects that the authors want to argue that CSDS alters hippocampal circuits specifically (e.g. increased ATP release here) which in turn activates microglia and leads to everything discussed. This is then the reason why hippocampal microglia are the focus. If this is the case, this needs to be more explicitly and clearly mentioned because as it currently reads like the study wants to claim something specifically unique about hippocampal microglia without much support [see also point #4 below]

RESPONSE: We appreciate the thoughtful criticism and suggestions provided by the reviewers. In our previous study, we had demonstrated the involvement of extracellular ATP-P2X7R-NLRP3 inflammasome signaling in hippocampal microglia in stress-induced depressive phenotypes and microglial activation ², as clearly stated in the Abstract and Introduction of this manuscript.

Building upon this foundation, our current study aimed to delve deeper into the intricate mechanisms underlying intracellular stress responses in microglia, with a particular focus on MAMs, to further elucidate this pathophysiological process. Thank you for highlighting this aspect, and we will ensure to clarify the rationale and objectives of our study accordingly.

[2] A second concern with the manuscript is the poor images. 1) All the IHC images are problematic because the cells can barely be seen. For example, in Fig. 1d, while there is roughly a 50% increase in Iba1 intensity according to the quantification, it is not clear that this is the case with the corresponding images. This is true for Fig. 2d and 7c as well. 2) Moreover, in these cases intensity of Iba1 is measured to be increased but it is not clear if this is because of an increased number of cells (i.e. density) or expression of Iba1. For Fig. 7c, the increase is referred as a density increase (line 287). This can/should be confirmed by ki67 staining as a proliferation marker. Especially for Fig. 2d, differences are not evident. Are different methods used in the analyses? No details on the Iba1 analysis is provided in the methods. This reviewer recommends that the authors present images with greater magnification so that the cells can be better visualized. 3) Moreover, it is not clear the usefulness of including the DAPI stain / merge image to indicate that the DG is examined. The manuscript is not focused on the DG but on hippocampus so given more space for visualization would be preferred that providing the DG context. Images like the zoomed in images of Fig. 6h are preferred, though the quality of those images can be improved. 5) In addition, in Fig. 2a images one can barely see anything. Fig. 3f is also barely visible because it is so small. Scale bars are barely visible in 4f. 6) Moreover, green/red colors should be avoided to accommodate color blind readers. For the EM images e.g. in Fig. 6d, it is not clear that the red arrows are pointing to the same things in the control and CSDS conditions. Careful organization and presentation of the Figures would improve the manuscript.

RESPONSE:

1) Thank you for your corrections and advice regarding the images. We have improved the quality of all image pixels in this version of the manuscript to enhance visibility and clarity.

2) Your comments regarding the Iba-1 optical density are insightful. While the optical density of Iba-1 cannot indicate the statement of microglia. To address this, we have analyzed microglial markers and microglial morphology on a cell-by-cell basis using Image J analysis. The results are now presented in Supplementary Figure 1D.

3) We appreciate your suggestion to include more detailed evaluation indicators and statistical methods related to microglial activation in the methods section. Additionally, we have expanded our analysis to include the evaluation of microglial activation in the CA1 regions of the hippocampus, in addition to the DG region. Detailed data are included in Figure S1H.

5) Thank you for your attention to detail. We have enlarged the relevant images in this version of the manuscript to improve visibility.

6) We have carefully considered your suggestion and adjusted the color scheme to avoid pairing red with green to accommodate color-blind readers. Furthermore, we have clarified the content indicated by the red arrow in the Figure legends for better interpretation by readers.

Thank you for your valuable feedback, and we believe these improvements will enhance the

overall quality and clarity of the manuscript.

[3] A third concern with the manuscript is the really complicated flow. This is a difficult manuscript to follow and needs to be read carefully (even multiple times) to be really understood. It feels that there are three parts to the paper. The first part (Fig. 1 & 2) seeks to establish microglial P2X7R signaling in the mediation of depressive like behaviors in a chronic social defeat stress model. First, it's not clear why this model was used over the chronic unpredictable mild stress model where P2X7R roles have been previously shown. Both models seem to generate depressive-like symptoms so why use a new model? The NLRP3 assembly features in the former model but doesn't seem to be relevant for the current model/story or even in the final model schematic. Why mention it? Pro-inflammatory microglial genes are mentioned in Fig. 1 following CSDS but the rest of the parts study does not really examine pro-inflammatory gene expression. These all serve to be distractions from understanding the focus of the study since they are not crucial to the central point(s) of the study.

RESPONSE: Thank you for your insightful comments and suggestions. Your feedback has prompted us to provide clearer explanations regarding the rationale behind our study design and the choice of animal models. Allow me to briefly explain here.

Firstly, while the chronic unpredictable mild stress (CUMS) model is indeed effective for studying depression, its main drawback lies in the lengthy paradigm required. Therefore, we opted to use the chronic social defeat stress (CSDS) model in this study due to its shorter duration.

Secondly, our previous investigations have demonstrated the significance of the extracellular ATP-P2X7R-NLRP3 inflammasome signaling pathway in CUMS-induced pro-inflammatory microglial activation and depressive phenotypes^{2,5}. Consequently, the aim of our current study is to elucidate the specific intracellular mechanisms underlying NLRP3 inflammasome assembly following P2X7R activation. Additionally, we seek to determine whether these mechanisms are implicated in CSDS-induced depressive phenotypes and microglial activation. Hence, it was necessary to validate our earlier findings obtained from the CUMS model in the context of the CSDS model.

As per your suggestion, we have de-emphasized this aspect of the results and relocated it to the Supplemental Figures to ensure that the main content remains unaffected. Furthermore, we have examined pro-inflammatory markers and NLRP3 inflammasome assembly in microglia both in vitro and in the hippocampus in vivo.

Also, although P2X7R KOs are available, they do not validate the antibody used on the KO mice to confirm the trustworthiness of that antibody in P2X7Rs. Could the authors do this?

RESPONSE: Thank you for raising this important point. We acknowledge the necessity of validating the antibody using P2X7R knockout (KO) mice to ensure its reliability and specificity.

However, for KO mice, we have validated their genotype using PCR mouse tail identification method (sFig.2a in the revised article), and the P2X7R antibody (sc-514962) used in experiments has been reported to be reliable in multiple literature^{6,7}.

Having confirmed microglial activation and P2X7R roles in the first part, the authors move to the second part to tease out intracellular mechanisms (Fig. 3-5) of ER-Mito dysfunction in ATP toxicity that presumably occurs in response to CSDS (though this is not shown directly or discussed). These studies are done in BV2 cells in vitro which have nothing specifically related to “hippocampal microglia”. This provides some disconnect with the previous part that the authors should try to limit. Moreover, there are concerns that in vitro microglia (and especially BV2 cells) do not recapitulate in vivo microglial phenotypes that the authors don’t address.

RESPONSE: Thank you for your valuable feedback and observations. We have carefully considered your concerns and have revised address the disconnect between the in vitro and in vivo experiments, as well as the limitations associated with BV2 cells.

In response to your comments, we have reorganized the results section to provide clarity and coherence in presenting our findings. Specifically, we have emphasized the induction of intracellular ER-MAMs-mitochondria stress responses in hippocampal microglia following CSDS (Fig. 1). Subsequently, we conducted in vitro experiments using primary microglia to validate our observations (Fig. 2), minimizing reliance on BV2 cells, which also been included in supplementary figures (Fig. S3).

We acknowledge the limitations associated with BV2 cells and have made efforts to prioritize experiments using primary microglia to better reflect in vivo microglial phenotypes. This approach strengthens the relevance and reliability of our findings and enhances the translational potential of our study.

The final part is a return to the in vivo CSDS model to confirm ER-mito damage that was observed in vitro (Fig. 6). Then they confirm that ER stress is sufficient to elicit the depression-like behaviors without CSDS (Fig. 7). This is great but the treatment is not specific to microglia so there is a serious gap in this flow with the story which the authors don’t address. Finally, the authors prevent the ER stress and mito dysfunction pharmacologically to improve ER-mito histology and depressive behaviors. But again this treatment is not specific to microglia. Given the commencement with P2X7R, it also feels like an incomplete circle to not test the drug in KOs as well.

RESPONSE: Indeed, it is acknowledged that ER stress inducers, inhibitors, or mitochondrial protectors lack specificity in targeting ER or mitochondria within microglia, thereby limiting their utility in elucidating the precise roles of ER stress and mitochondrial damage in microglial function within the context of CSDS-induced depressive phenotypes. Consequently, the relevant findings pertaining to these treatments have been omitted from the current manuscript iteration.

Instead, we employed Cx3cr1^{CreER/+}/Hspa9^{fllox/+} mice, which were bred in the previous year, to investigate the involvement of MAMs and mitochondrial alterations specifically within microglia in the context of CSDS-induced depressive phenotypes and microglial activation. The pertinent results derived from these experiments have been incorporated into the present manuscript (Fig.4) for comprehensive elucidation.

Therefore, to this reviewer, while the template of the story is nice: the combination of in vivo and in vitro approaches etc, the links made between each of these three parts are tenuous at times. Independently each part could stand but together the authors need to do a better job in making the appropriate connections and claims and at least discussing their study limitations. It is the view of this reviewer that simplifying the flow to be more focused will make more compelling manuscript. Some panels of the current main figures can be moved to Supplemental Figures to allow for more space to explain and visualize some of the critical (especially images) panels [see also point #2]

RESPONSE: We sincerely appreciate the valuable insights provided by the reviewers, which have prompted us to reconsider the structure and coherence of our manuscript. Upon reflection, we acknowledge that the initial presentation of results may have led to a less cohesive narrative. Consequently, we have reorganized the manuscript to enhance clarity and focus.

Our revised manuscript begins by establishing the impact of stress on intracellular responses in microglia, notably highlighting ER stress, structural anomalies in MAMs, and ensuing mitochondrial dysfunction. Concurrently, we examine hippocampal microglial activation and the manifestation of depressive phenotypes. Subsequent in vitro investigations demonstrate the induction of similar intracellular stress responses in primary microglia, notably mediated through P2X7R upon exposure to high doses of ATP (1mM).

Of particular significance, we elucidate the role of IP3R-GRP75-VDAC1 aggregation within MAMs as a pivotal intermediary response, triggering NLRP3 inflammasome assembly via the eATP-P2X7R signaling pathway. Complementing these findings, in vivo studies underscore the therapeutic potential of microglia-specific GRP75 knockdown, effectively attenuating pathological changes in depressive phenotypes and microglial activation induced by social defeat stress.

Lastly, employing P2X7 knockout mice, we delineate the pivotal role of P2X7R in mediating stress-induced intracellular responses within microglia, including alterations in MAMs structure and function. Collectively, these experiments offer comprehensive insights into the intricate mechanisms through which stress stimuli precipitate depressive phenotypes and inflammatory microglial activation within the hippocampus.

Considering these revisions, we believe our manuscript now presents a more streamlined and coherent narrative, elucidating the interplay between stress, microglial dynamics, and depressive phenotypes. We have also taken note of the suggestion regarding the relocation of certain panels to Supplemental Figures, thereby enhancing the clarity of critical visualizations while optimizing space for detailed explanations.

[4] A final major concern is the language of the manuscript. While much of the language is understandable, some of it makes for very difficult reading. For example, the Abstract was very difficult to understand for this reviewer. "proposed" in line 31 should be better replaced with something like "investigated". The results were simply stated [lines 31-41] and it was difficult to initially grasp whether the authors performed these experiments and generated these results, or these are background facts. Descriptors like "we find/found that...", "we show/showed that...", "we discover/discovered that...". In line 40, "especially" in microglia is mentioned but this is

somewhat misleading because they only study microglia throughout the manuscript. The use of “while” on line 109 is quite confusing as it kind of sets off a possible contrast but the two ideas are not contrasting. The authors should incorporate a careful and thorough proof-reading / editing of the manuscript in a subsequent submission especially with the help of (an) English-proficient editor(s).

RESPONSE: We greatly appreciate the reviewer's feedback regarding the language of the manuscript. Following your recommendation, we have undertaken thorough language editing and refinement with the assistance of an English-proficient editor.

Minor Concerns

[1] Sample sizes used for in vitro experiments were unclear e.g. “n = 5 per condition” 5 cells? 5 experiments? Fields of view? How many cells per experiment/fields of view etc? For Fig. 3a-c; or “n = 4” for Fig. 3d 4 what? Cells? experiments? How many cells? etc. The authors should provide these types of information throughout the Figures.

RESPONSE: The figure legends have been revised to include specific details regarding sample sizes used for in vitro experiments, addressing the concern raised regarding clarity.

[2] Fig. 2d should have data for the basal florescence without CSDS to confirm that P2RY12 does not regulate basal Iba1 expression.

RESPONSE: We are grateful for your attention to detail and constructive suggestion regarding Fig. 2d. In response to your recommendation, we have included fluorescent contrast images in the updated version of the manuscript to provide data for the basal fluorescence without CSDS (sFig.2b in the revised article).

Reviewer #3:

In their manuscript Zhang et al have attempted to highlight the role of ER-mitochondria tethering in mouse models of depression. The authors have also performed a wide array of experiments in cell lines and primary microglial cells to support their in vivo findings and have made an effort to provide a more sophisticated insight of the underlying mechanisms.

I regret to write in this review that this paper has many major issues on every level.

I think that this paper suffers from poor writing and in several cases poor interpretation and analysis of the data as well as technique application. Although I will give some examples to justify my opinion, I will not go through every detail. I sincerely hope that the authors find the following criticism constructive.

RESPONSE: We appreciate the detailed feedback provided in your review. Your insights will undoubtedly contribute to the refinement of our manuscript. In response to your concerns, we have conducted additional experiments to bridge various sections of the manuscript and have thoroughly revised the paper to enhance its logical coherence and clarity.

Firstly, in terms of writing, the paper lacks logical structure. Already the abstract, for someone who did not carefully read the entire manuscript, is very complex and disconnected. 1) It is even hard to understand the main point of the paper when psychiatric disorders are presented with mitochondrial impairment as a culprit (reference 22) and the authors focus on microglial cells and not neurons.

RESPONSE: Thank you for your feedback. Our emphasis on microglial cells and mitochondrial impairment in psychiatric disorders stems from the recognition that while neurons have been extensively studied in this context, the role of mitochondrial dysfunction in glial cells, particularly microglia, has been relatively overlooked until recently.

While neurons heavily rely on mitochondrial function for energy production, emerging evidence suggests that mitochondrial metabolism and signaling pathways also play crucial roles in regulating glial cell function, including microglia. This shift in focus highlights the potential significance of mitochondrial dysfunction in glial cells, particularly in the context of psychiatric disorders such as depression, where neuroinflammation is a key contributor.

Mitochondrial dysfunction in microglia has been implicated in triggering inflammatory responses, exacerbating neuroinflammatory processes associated with psychiatric disorders. Investigating the role of mitochondrial dysfunction in microglia represents a promising avenue for advancing our understanding of the complex mechanisms underlying these disorders.

We appreciate your insights and are committed to further exploring the interplay between mitochondrial dysfunction in microglia and psychiatric disorders to identify potential therapeutic targets.

2) Used experimental approaches are not justified (e.g., why did the authors choose to perform an experiment in which they placed aCSF in mice? Or why was a sub-CSDS used? Why were A839977 and BzATP chosen?) and the presented points do not follow any obvious rationale (e.g., why did they look if Ca²⁺ is gated into mitochondria (lines 165-166)?

RESPONSE: Thank you for your detailed review and comments. We have provided in the revised manuscript our full rationale for adopting these experimental methods, and the basis for conducting these studies.

This study endeavors to elucidate the specific intracellular mechanisms underlying NLRP3 inflammasome assembly upon P2X7R activation, building upon our prior findings that highlighted the pivotal role of the eATP-P2X7R-NLRP3 inflammasome signaling pathway in CUMS-induced pro-inflammatory microglial activation and depressive phenotypes^{1,2}.

To validate our earlier results obtained from the CUMS model, we sought to investigate whether these intracellular mechanisms also contribute to CSDS-induced depressive phenotypes and microglial activation. Hence, we utilized the P2X7R antagonist A839977 and P2X7R agonist BzATP, with aCSF serving as a vehicle control.

Additionally, we incorporated sub-CSDS to explore the potential impact of ER stress inducers on stress susceptibility in animals. Following feedback from another reviewer, this section has been

omitted in the revised version of our manuscript.

As we understand, the P2X7 receptor is a ligand-gated cation channel activated upon binding with ATP. It facilitates the passage of various cations, including sodium, potassium, and calcium ions. The endoplasmic reticulum, serving as an intracellular calcium reservoir, exhibits sensitivity to alterations in intracellular cation concentrations. Through the IP3R3-VDAC-GRP75 complex in MAMs, the endoplasmic reticulum can transfer calcium to mitochondria, a process associated with mitochondrial calcium overload and subsequent damage. Moreover, by-products of mitochondrial damage, such as released mtDNA and reactive oxygen species, have the potential to directly induce the activation of the NLRP3 inflammasome. Consequently, we posit that ER stress, alterations in MAMs, and mitochondrial damage may constitute intracellular stresses triggering mitochondrial dysfunction through the activation of P2X7 receptors (cationic currents). In line with this hypothesis, we endeavored to assess the concentrations of endoplasmic reticulum and mitochondrial calcium ions in microglia following ATP stimulation.

Thank you for your valuable feedback, and we are committed to refining our manuscript to address these concerns and enhance the clarity and coherence of our study.

3) *How are the data presented in the 1st section of the results “further confirm the function of P2X7Rs” (line 105)?*

RESPONSE: In the last section of results in revised manuscript, we stated the involvement of P2X7R in the formation of the IP3R3-GRP75-VDAC1 complex within microglia in response to eATP, as well as its impact on the changes in MAMs within hippocampal microglia and the subsequent development of depression-like behaviors triggered by CSDS. Our findings, based on primary microglia from P2X7R^{-/-} mice, demonstrated that the deletion of P2X7R prevented the enrichment of ER-mitochondria contacts and the formation of the IP3R3-GRP75-VDAC1 complex (Fig. 5A-B) induced by a high concentration of eATP. Similar results were obtained using siRNA-treated BV2 cells (Fig. S5A-B). Additionally, the structural changes in MAMs in hippocampal microglia (Fig. 5C) and depression-like behaviors (Fig. S2E) caused by exposure to CSDS were attenuated in P2X7R^{-/-} mice.

These experiments collectively provide insights into the functional involvement of P2X7Rs in microglial dynamics, particularly in the context of depressive behaviors and intracellular interactions following stress induction.

The article needs to be completely restructured to connect its various aspects and make it easier to follow. The working hypothesis needs to become clearer and more aligned with the data and all results should be part of a one bigger connected story. In the present form of the article, the findings seem unconnected and irrelevant to one another while the authors still jump into rather daring conclusions (lines 174-175, lines 221-222, lines 230-233 etc). It is quite clear that in an effort to connect so many different concepts (depressive-like phenotypes, microglia activation, inflammatory markers and complexes, P2X7R receptors, ER-mitochondria tethering and Ca²⁺ dynamics) the logical connections have been majorly neglected. Perhaps removing or condensing some of the results will make the points of the work sharper and in focus.

RESPONSE: Thank you for your valuable suggestion. We acknowledge the challenge of connecting numerous concepts within the scope of our study while maintaining logical coherence. It is evident that attempting to address all aspects simultaneously has led to a lack of clear connections between findings.

In response to your feedback and that of other reviewers, we have undertaken a comprehensive restructuring of the article. By splitting and reorganizing the content, we aim to streamline the presentation of our experimental results and conclusions. This revised approach will allow for a more focused and coherent narrative, enabling readers to better grasp the interconnectedness of our findings.

We appreciate your insightful critique and are committed to enhancing the clarity and cohesion of our manuscript.

Despite the fact that major restructuring of the paper would comprise a significant improvement, I still think that in several cases the data analysis needs extensive implementations. 1) For example, in many cases it is unclear how many independent biological replicates have been used (e.g., Figures 1d, 3a, 3b, 3c, 3f). 2) Also, in some other instances the chosen statistical tests were parametric when only 3 points/group were compared and a non-parametric test would be more appropriate (e.g., figures 1d and 2d). 3) Additionally, in some figures (e.g., 1d and 2d) the control bars do not have any error bars that is not only biologically impossible, but also statistically incorrect while at the same time it biases any applied statistical test to give significant results. 4) Another sign that the authors should consult with an expert regarding their data analysis, is the Venn diagram in Figure 1f. The data on this diagram have no biological or statistical meaning based on what is described in the text. In conclusion, 5) this paper lacks on many occasions' proper statistical analysis. This is important because correct statistical analysis or addition of more data points might change the significance of the data, ergo their interpretation in this particular work.

RESPONSE: Thank you for your thorough review and valuable feedback, which have significantly contributed to our understanding of the data analysis shortcomings in our manuscript. We have taken your points into careful consideration and implemented the following measures to address these concerns.

We have provided detailed descriptions of the number of independent biological replicates used in Figures 1d (sFig.1c in the revised article), 3a, 3b, 3c, and 3f (Fig 2.f, g,d in the revised article) within the Figure Legends section of the revised article. The corresponding legends are as follows: sFig.1c. Representative immunofluorescent images (left panel) and semi-quantitative analysis (right panel) of Iba-1 fluorescence intensity of hippocampal microglia in dentate gyrus (DG) from control and CSDS mice. Iba-1 (green), DAPI (blue). Scale bar: 20 μ m; Student's *t*-test. Data are expressed as mean \pm SEM (n = 4). ****p* < 0.001.

Fig.2f. Representative fluorescence images, 3D thermograms (left panel) and quantitative analysis (right panel) of Fluo-4 in microglia. Scale bar: 20 μ m. One-way ANOVA with Dunnett's multiple comparisons test. Data are expressed as mean \pm SEM (n = 5 per condition). ****p* < 0.001, vs. Ctrl.

Fig.2g. Representative fluorescence images (left panel) and quantitative analysis (right panel) of

Mag-Fluo-4 and Rhod-2 in microglia. Scale bar: 20 μm . One-way ANOVA with Dunnett's multiple comparisons test. Data are expressed as mean \pm SEM (n = 4 per condition). * $p < 0.05$, ** $p < 0.01$, *** $p < 0.001$, vs. Ctrl.

Fig.2d. Representative confocal images (left panel) and fluorescence intensity at the line of the arrow in left images (right panel) of the intracellular localization of lysosome (Lyso-Tracker, red), mitochondria (Mito-Tracker, green) and nucleus (Hoechst, blue) in microglia after 2h or 12h of incubation with ATP or TG. Scale bar: 20 μm .

Non-parametric statistical tests have been employed for data analysis where applicable, particularly in Figures 1d (sFig.1c in the revised article) and 2d (sFig.2b in the revised article), where only three data points per group were compared.

We apologize for the absence of error bars in the control bars of Figures 1d (sFig.1c in the revised article) and 2d (sFig.2b in the revised article). We have rectified this oversight and ensured that error bars are appropriately included in the revised figures.

In response to your observation regarding the Venn diagram in Figure 1f, we agree that it lacked biological and statistical meaning based on the description in the text. Consequently, the Venn diagram has been removed from the revised version of the manuscript.

To ensure proper statistical analysis, we have consulted with a statistician expert and reanalyzed the data using appropriate methods. The results statements have been revised accordingly to accurately reflect the findings.

We appreciate your diligence in highlighting these issues and are committed to upholding the highest standards of data analysis and interpretation in our research.

Finally, I strongly disagree with some of the experimental approaches or selected experimental methods. 1) The data presented in figures 2b and 2c are a good example to support my point. Although it is not explained in the text (the lack of sufficient explanation of the data is spread throughout the manuscript), it seems that the aCSF+CSDS group is set as a control and based on its performance the authors have concluded that A839977+CSDS mice have impeded development of depressive-like phenotypes. However, if one compares the performance scores of this figure with the ones of figure 1c, it is clear that A839977+CSDS have the same score with CSDS animals for SIT, which is around 120. This clearly indicates that figure 2c lacks vital controls and the substitution of a proper control with aCSF+CSDS affects the interpretation of an experiment that is central for the first part of the study. 2) Another main issue is the use of an ER stain and mitotracker to determine the proximity of ER and mitochondria, an assay that has been used extensively in the paper (e.g. Figures 4 and 5) and supports several major claims although the method is inappropriate. The 2 accepted methods to determine ER-mitochondria proximity are TEM and PLA. Curiously, the authors did perform these assays in some instances (Figure 4b and Figure 4f, respectively) but failed to apply them in others while at the same time they based many of their conclusions on the results given by an inappropriate method.

RESPONSE: Thank you for raising these concerns. It appears that the interpretation of the data in Figures 2b and 2c may have been influenced by the selection of the control group. Upon

reevaluation, it seems that the aCSF+CSDS group served as a control, potentially impacting the conclusions drawn regarding the effects of A839977+CSDS mice on the development of depressive-like phenotypes. We acknowledge the importance of proper controls and have subsequently established two control groups in future trials to assess the impact of the P2X7R antagonist, A839977, on CSDS-induced depressive-like behavior: the Control group, Control+ A839977 group, which has been included in Figure S2D.

Regarding the use of ER stain and MitoTracker to assess ER-mitochondria proximity, we acknowledge your point about the appropriateness of the method. While we did perform TEM and PLA assays in some instances (Figure 4B and Figure 4H), we recognize the importance of consistency in methodology. Hence, we have incorporated TEM and PLA assays both *in vivo* and *in vitro* to accurately determine ER-mitochondria proximity, as per your suggestion.

Your insights are invaluable, and we are committed to upholding rigorous scientific standards in our experimental approaches and data interpretation. Thank you for bringing these issues to our attention.

Overall, the authors should take this opportunity and consider the above comments. Improving the quality of their data as well as restructuring their story has the potential of bringing forward impactful science. However, I am afraid that, in its present form, this work is not suitable for publication. I wish I could deliver a more positive opinion and once again, I sincerely hope that the authors find this criticism helpful.

RESPONSE: Thank you for your recognition and encouragement. We have made significant efforts to address your comments by constructing transgenic animals, conducting additional *in vitro* and *ex vivo* experiments, restructuring the results, and rewriting the paper. We hope that these revisions will meet your approval in the revised manuscript version.

References:

1. Chen LQ, *et al.* Asymmetric activation of microglia in the hippocampus drives anxiodepressive consequences of trigeminal neuralgia in rodents. *Br J Pharmacol* **180**, 1090-1113 (2023).
2. Yue N, *et al.* Activation of P2X7 receptor and NLRP3 inflammasome assembly in hippocampal glial cells mediates chronic stress-induced depressive-like behaviors. *J Neuroinflammation* **14**, 102 (2017).
3. Iwata M, *et al.* Psychological Stress Activates the Inflammasome via Release of Adenosine Triphosphate and Stimulation of the Purinergic Type 2X7 Receptor. *Biol Psychiatry* **80**, 12-22 (2016).
4. Dang R, *et al.* Edaravone ameliorates depressive and anxiety-like behaviors via Sirt1/Nrf2/HO-1/Gpx4 pathway. *J Neuroinflammation* **19**, 41 (2022).
5. Wang YL, *et al.* Microglial activation mediates chronic mild stress-induced depressive- and anxiety-like behavior in adult rats. *J Neuroinflammation* **15**, 21 (2018).

6. Li LH, *et al.* Critical Role for the NLRP3 Inflammasome in Mediating IL-1 β Production in *Shigella sonnei*-Infected Macrophages. *Front Immunol* **11**, 1115 (2020).
7. Wei C, *et al.* The NLRP3 inflammasome regulates corneal allograft rejection through enhanced phosphorylation of STAT3. *Am J Transplant* **20**, 3354-3366 (2020).

REVIEWERS' COMMENTS

Reviewer #1 (Remarks to the Author):

Thanks for the revised version. The authors have addressed all my comments.

Reviewer #2 (Remarks to the Author):

Review of "Augmented Microglial Endoplasmic Reticulum-Mitochondria Contacts Mediate Chronic Social Defeat Stress-Induced Depression"

This is a revised manuscript by Zhang et al. and it is a much improved manuscript. A few (mostly minor) issues still remain as highlighted below for this reviewer before it can be suitable for acceptance.

Minor problems.

1. Some data is presented as increased florescence, and it is not clear whether this is simply an increase in the expression of the marker or an increase in the density of cells. The authors should provide quantification to ascertain increase density for e.g. Fig. Supp. Fig. 1c, h, i and 2b.
2. I response to point #3 of the first submission, this reviewer suggest a validation of the antibody using P2RX7 Kos but it seems the authors misunderstood this to be a request to validate the KO mice to be truly KO. The request still remains valid and unaddressed. Please, validate that the P2RY12 antibody in use with the KO tissue i.e. show that P2RX7 is detectable in the WT but not the KO tissue.
3. The word "statement" on line 55 should be replaced with the word "status".
4. There should be "of" in between "development" and "various diseases" on line 63.
5. The word "impeded" would be better replaced with the word "prevented" in line 163.
6. There is a slight mis-match for Supp. Fig 1f / g where the legends/figure versus the main text (lines 119 and 120) differ between the mention of RT-PCR or RNAseq I . Please, rectify.
7. There should be a mention of Thapsigargin and what it does in the text descriptions for Supp Fig 3 in section 2 of the Results section.
8. In the control mice data for Fig. 4d in the SIT, is there a significant difference?
9. To appropriately interpret the data for Fig. 4e-h, the authors should also show the non CSDS data.
10. Figures: The figures should be improved as:
 - a. the text of Fig. 1c/d cannot be read clearly;
 - b. Data in Fig. 1e/f should be quantified to show an increase;
 - c. Images in Fig. 1 in general and especially Fig. 1g are small;
 - d. Images in Fig. 2e and 3e are so small and can barely be seen;
 - e. Images in Fig. 3f already seem to show high DCF expression in the ATP group to start with. Perhaps replace this image for a more representative image.
 - f. Please, show the graph for Casp1 and Asc rather than the blot images for Fig. 3h

Reviewer comments are in *italics*, and author responses are in regular text, highlighted in blue.

REVIEWERS' COMMENTS

Reviewer #1 (Remarks to the Author):

Thanks for the revised version. The authors have addressed all my comments.

RESPONSE: I appreciate your acknowledgment of the revised version.

Reviewer #2 (Remarks to the Author):

Review of “Augmented Microglial Endoplasmic Reticulum-Mitochondria Contacts Mediate Chronic Social Defeat Stress-Induced Depression”

This is a revised manuscript by Zhang et al. and it is a much-improved manuscript. A few (mostly minor) issues still remain as highlighted below for this reviewer before it can be suitable for acceptance.

Minor problems.

1. Some data is presented as increased florescence, and it is not clear whether this is simply an increase in the expression of the marker or an increase in the density of cells. The authors should provide quantification to ascertain increase density for e.g. Fig. Supp. Fig. 1c, h, i and 2b.

RESPONSE: Thank you for your valuable suggestion. We have re-quantified and statistically analyzed fluorescence expression in these images.

2. I response to point #3 of the first submission, this reviewer suggests a validation of the antibody using P2RX7 Kos but it seems the authors misunderstood this to be a request to validate the KO mice to be truly KO. The request still remains valid and unaddressed. Please, validate that the P2RY12 antibody in use with the KO tissue i.e. show that P2RX7 is detectable in the WT but not the KO tissue.

RESPONSE: We apologize for the misunderstanding. Following your recommendation, we proceeded with antibody validation using brain tissue samples obtained from both P2RX7 knockout (KO) and wild-type (WT) mice. However, the Western blot analysis unveiled the presence of detectable levels of P2X7R in the brain of KO mice, as depicted in Figure 1a. This observation aligns with the information provided on the Jackson Laboratory website regarding P2RX7 KO mice (<https://www.jax.org/strain/005576>), which indicates that low levels of the 13b isoform of P2X7R and a C-terminal truncated variant transcript can still be discerned in the brains of KO mice.

As per the documentation from the Jackson Laboratory, this particular transgenic mouse model is engineered to disrupt the carboxy-terminal coding region of the target gene by employing a targeting vector that contains a neomycin resistance gene driven by the mouse

phosphoglycerate kinase promoter. Consequently, in this transgenic mouse, the sequence of the P2X7 gene (which encodes amino acids 506 to 532) was replaced by the insertion of a neomycin selection cassette. The specific antigen targeted by the antibody sc-514962, which we used is a mouse monoclonal antibody specific for an epitope mapping between amino acids 81-106 within an internal region of P2X7 of human origin. Therefore, this antibody still detects the 13b isoform and the C-terminal truncated variant transcript of P2X7R expressed at low levels in KO mouse brain.

To validate antibody specificity, we consulted the expression profile of P2RX7 (<https://www.proteinatlas.org/ENSG00000089041-P2RX7>), which indicated negligible expression in the LoVo cell line. Consequently, we evaluated the antibody's specificity using LoVo cells, as depicted in Figure 1b. The results suggest that the antibody maintains a high degree of specificity.

Figure 1. P2X7R antibody validation. a. Western-blot validation of hippocampal tissue in P2RX7 knockout (KO) and wild-type (WT) mice; b. Antibody validation using human colon cancer LoVo cell lines.

3. The word “statement” on line 55 should be replaced with the word “status”.

RESPONSE: Thank you for your suggestion. We have made corrections in the main text.

4. There should be “of” in between “development” and “various diseases” on line 63.

RESPONSE: I appreciate your suggestion. We have made corrections in the main text.

5. The word “impeded” would be better replaced with the word “prevented” in line 163.

RESPONSE: Thanks for your suggestion. We have made corrections in the main text.

6. There is a slight mis-match for Supp. Fig 1f/ g where the legends/figure versus the main text (lines 119 and 120) differ between the mention of RT-PCR or RNAseq. Please, rectify.

RESPONSE: Please accept my sincerest apologies for my mistake. We have made corrections in the main text.

7. There should be a mention of Thapsigargin and what it does in the text descriptions for Supp Fig 3 in section 2 of the Results section.

RESPONSE: We sincerely appreciate your constructive comments. The specific functions have been described in the main text, as follows:

“The results demonstrated that after 2 hours of ATP (1mM) or 12 hours of ER stress inducer Thapsigargin (TG, 1μM, used as a positive control, which raises cytosolic calcium concentration by blocking the ability of the cell to pump calcium into the sarcoplasmic reticulum and ER) treatment resulted in an augmentation of ER-mitochondria contacts, as evidenced by transmission electron microscopy experiments (Fig. 2a and Supplementary Fig. 3a) and...”

8. In the control mice data for Fig. 4d in the SIT, is there a significant difference?

RESPONSE: No, the results of the Two-way with Tukey multiple comparison test showed that there was no significant difference between the two groups with $P > 0.999$. The specific analysis results are as follows:

Tukey's multiple comparisons test	Mean Diff.	95.00% CI of diff.	Significant?	Summary	Adjusted P Value
Cx3cr1 ^{CreER} :Control vs. Cx3cr1 ^{CreER} :CSDS	0.7730	-0.4033 to 1.949	No	ns	0.2851
Cx3cr1 ^{CreER} :Control vs. Cx3cr1 ^{CreER} /Hspa9 ^{fllox/+} :Control	0.001763	-1.175 to 1.178	No	ns	>0.9999
Cx3cr1 ^{CreER} :Control vs. Cx3cr1 ^{CreER} /Hspa9 ^{fllox/+} :CSDS	-0.2524	-1.429 to 0.9239	No	ns	0.9307
Cx3cr1 ^{CreER} :CSDS vs. Cx3cr1 ^{CreER} /Hspa9 ^{fllox/+} :Control	-0.7712	-1.948 to 0.4051	No	ns	0.2869
Cx3cr1 ^{CreER} :CSDS vs. Cx3cr1 ^{CreER} /Hspa9 ^{fllox/+} :CSDS	-1.025	-2.202 to 0.1510	No	ns	0.1015
Cx3cr1 ^{CreER} /Hspa9 ^{fllox/+} :Control vs. Cx3cr1 ^{CreER} /Hspa9 ^{fllox/+} :CSDS	-0.2541	-1.430 to 0.9222	No	ns	0.9294

9. To appropriately interpret the data for Fig. 4e-h, the authors should also show the non CSDS data.

RESPONSE: Thank you for insightful comments and suggestions. We supplemented the control group data involved in the experiment in Figure 4e-h and presented it in Supplementary Fig. 4d-g.

10. Figures: The figures should be improved as:

- the text of Fig. 1c/d cannot be read clearly;
- Data in Fig. 1e/f should be quantified to show an increase;
- Images in Fig. 1 in general and especially Fig. 1g are small;
- Images in Fig. 2e and 3e are so small and can barely be seen;
- Images in Fig. 3f already seem to show high DCF expression in the ATP group to start with. Perhaps replace this image for a more representative image.
- Please, show the graph for Casp1 and Asc rather than the blot images for Fig. 3h

RESPONSE: Thank you for your corrections and advice regarding the images. We've taken the following actions to address your feedback:

- Enlarged the font of the text in Fig. 1c/d for improved readability.
- Added a data statistics chart for Fig. 1e/f to quantify the data and demonstrate the increase.
- Enlarged the images in Fig. 1, Fig. 2e, and 3e to enhance visibility.
- Adjusted the size of images in Fig. 2e and 3e for better clarity.
- Corrected the order of the images and replaced Fig. 3f with a more representative image to avoid confusion.

f. Included Caspase-1 and ASC protein statistics in Fig. 3h, as per your constructive suggestion.